# Dissecting maternal and fetal genetic effects underlying the associations between maternal phenotypes, birth outcomes, and adult phenotypes: A mendelian-randomization and haplotype-based genetic score analysis in 10,734 mother–infant pairs

Jing Chen[1], Jonas Bacelis[2,3], Pol Sole-Navais[2], Amit Srivastava[4,5], Julius Juodakis[2], Amy Rouse[5], Mikko Hallman[6], Kari Teramo[7], Mads Melbye[8,9,10], Bjarke Feenstra[8], Rachel M. Freathy[11], George Davey Smith[12,13,14], Deborah A. Lawlor[12,13,14], Jeffrey C. Murray[15], Scott M. Williams[16], Bo Jacobsson[2,17], Louis J. Muglia[4,5]*, Ge Zhang[4,5]*

1 Division of Biomedical Informatics, Cincinnati Children's Hospital Medical Center, Department of Pediatrics, University of Cincinnati College of Medicine, Cincinnati, Ohio, United States of America, 2 Department of Obstetrics and Gynecology, Institute of Clinical Sciences, Sahlgrenska Academy, University of Gothenburg, Gothenburg, Sweden, 3 Region Västra Götaland, Sahlgrenska University Hospital, Department of Obstetrics and Gynecology, Gothenburg, Sweden, 4 Division of Human Genetics, Cincinnati Children's Hospital Medical Center, Department of Pediatrics, University of Cincinnati College of Medicine, Cincinnati, Ohio, United States of America, 5 Center for Prevention of Preterm Birth, Perinatal Institute and March of Dimes Prematurity Research Center Ohio Collaborative, Cincinnati Children's Hospital Medical Center, Department of Pediatrics, University of Cincinnati College of Medicine, Cincinnati, Ohio, United States of America, 6 PEDEGO Research Unit and Medical Research Center Oulu, University of Oulu and Department of Children and Adolescents, Oulu University Hospital, Oulu, Finland, 7 Obstetrics and Gynecology, University of Helsinki and Helsinki University Hospital, Helsinki, Finland, 8 Department of Epidemiology Research, Statens Serum Institut, Copenhagen, Denmark, 9 Department of Clinical Medicine, University of Copenhagen, Copenhagen, Denmark, 10 Department of Medicine, Stanford University School of Medicine, Stanford, California, United States of America, 11 Institute of Biomedical and Clinical Science, College of Medicine and Health, University of Exeter, Exeter, United Kingdom, 12 MRC Integrative Epidemiology Unit at the University of Bristol, Bristol, United Kingdom, 13 Population Health Science, Bristol Medical School, University of Bristol, Bristol, United Kingdom, 14 Bristol NIHR Biomedical Research Centre, United Kingdom, 15 Department of Pediatrics, University of Iowa, Iowa City, Iowa, United States of America, 16 Department of Population and Quantitative Health Sciences, Case Western Reserve University School of Medicine, Cleveland, Ohio, United States of America, 17 Department of Genetics and Bioinformatics, Domain of Health Data and Digitalisation, Institute of Public Health, Oslo, Norway

* ge.zhang@cchmc.org (GZ); louis.muglia@cchmc.org (LJM)

## Abstract

### Background

Many maternal traits are associated with a neonate's gestational duration, birth weight, and birth length. These birth outcomes are subsequently associated with late-onset health conditions. The causal mechanisms and the relative contributions of maternal and fetal genetic effects behind these observed associations are unresolved.

**Data Availability Statement:** The data used in this study are available to other researchers. In order to respect and protect the interests and privacy of the research participants, the access to the individual-level phenotype and genotype data requires submitting applications to and approval by the corresponding entities who are in charge of the distribution of the data sets (e.g., FIN, ALSPAC, FIN, MoBa, and dbGaP). This is to ensure that the proposed study aims are consistent with the informed consent under which the data or samples were collected and appropriate data safety and security measures are in place to protect against data breach and unauthorized use. Individual-level phenotype and genotype data from the Finnish (Helsinki) birth cohort are available through the March of Dimes Prematurity Research Center Ohio Collaborative (http://prematurityresearch.org/ohiocollaborative/), and access will be approved by the Leadership Committee through its director of operations, Joanne Chappell (joanne.chappell@cchmc.org). ALSPAC data are available to scientists on request to the ALSPAC Executive Committee (ALSPAC-exec@bristol.ac.uk) or via website (http://www.bristol.ac.uk/alspac/researchers/access/), which also provides full details and distributions of the ALSPAC study variables. The detailed policy of data sharing can be found in the ALSPAC data management plan (http://www.bristol.ac.uk/alspac/researchers/data-access/documents/alspac-data-management-plan.pdf). MoBa data is available to researchers and research groups at both the Norwegian Institute of Public Health and other research institutions nationally and internationally. The research must adhere to the aims of MoBa and the participants' given consent. All use of data and biological material from MoBa is subject to Norwegian legislation. Terms for applying for access to data and links to the application form and information can be found at https://www.fhi.no/en/studies/moba/for-forskere-artikler/research-and-data-access/. Access to the DNBC (phs000103.v1.p1), HAPO (phs000096.v4.p1), and GPN (phs000714.v1.p1) individual-level phenotype and genetic data can be obtained through dbGaP Authorized Access portal (https://dbgap.ncbi.nlm.nih.gov/dbgap/aa/wga.cgi?page=login). The informed consent under which the data or samples were collected is the basis for determining the appropriateness of sharing data through unrestricted-access databases or NIH-designated controlled-access data repositories.

**Funding:** This work is supported by a grant from the Burroughs Wellcome Fund (10172896) and a grant from the Cincinnati Children's Hospital Medical Center (GAP/RIP) to GZ and a grant from

## Methods and findings

Based on 10,734 mother–infant duos of European ancestry from the UK, Northern Europe, Australia, and North America, we constructed haplotype genetic scores using single-nucleotide polymorphisms (SNPs) known to be associated with adult height, body mass index (BMI), blood pressure (BP), fasting plasma glucose (FPG), and type 2 diabetes (T2D). Using these scores as genetic instruments, we estimated the maternal and fetal genetic effects underlying the observed associations between maternal phenotypes and pregnancy outcomes. We also used infant-specific birth weight genetic scores as instrument and examined the effects of fetal growth on pregnancy outcomes, maternal BP, and glucose levels during pregnancy. The maternal nontransmitted haplotype score for height was significantly associated with gestational duration ($p = 2.2 \times 10^{-4}$). Both maternal and paternal transmitted height haplotype scores were highly significantly associated with birth weight and length ($p < 1 \times 10^{-17}$). The maternal transmitted BMI scores were associated with birth weight with a significant maternal effect ($p = 1.6 \times 10^{-4}$). Both maternal and paternal transmitted BP scores were negatively associated with birth weight with a significant fetal effect ($p = 9.4 \times 10^{-3}$), whereas BP alleles were significantly associated with gestational duration and pre-term birth through maternal effects ($p = 3.3 \times 10^{-2}$ and $p = 4.5 \times 10^{-3}$, respectively). The nontransmitted haplotype score for FPG was strongly associated with birth weight ($p = 4.7 \times 10^{-6}$); however, the glucose-increasing alleles in the fetus were associated with reduced birth weight through a fetal effect ($p = 2.2 \times 10^{-3}$). The haplotype scores for T2D were associated with birth weight in a similar way but with a weaker maternal effect ($p = 6.4 \times 10^{-3}$) and a stronger fetal effect ($p = 1.3 \times 10^{-5}$). The paternal transmitted birth weight score was significantly associated with reduced gestational duration ($p = 1.8 \times 10^{-4}$) and increased maternal systolic BP during pregnancy ($p = 2.2 \times 10^{-2}$). The major limitations of the study include missing and heterogenous phenotype data in some data sets and different instrumental strength of genetic scores for different phenotypic traits.

## Conclusions

We found that both maternal height and fetal growth are important factors in shaping the duration of gestation: genetically elevated maternal height is associated with longer gestational duration, whereas alleles that increase fetal growth are associated with shorter gestational duration. Fetal growth is influenced by both maternal and fetal effects and can reciprocally influence maternal phenotypes: taller maternal stature, higher maternal BMI, and higher maternal blood glucose are associated with larger birth size through maternal effects; in the fetus, the height- and metabolic-risk–increasing alleles are associated with increased and decreased birth size, respectively; alleles raising birth weight in the fetus are associated with shorter gestational duration and higher maternal BP. These maternal and fetal genetic effects may explain the observed associations between the studied maternal phenotypes and birth outcomes, as well as the life-course associations between these birth outcomes and adult phenotypes.

the March of Dimes (22-FY17-889) and a grant from the Bill and Melinda Gates Foundation (OPP1175128) to LJM and GZ. The Norwegian Mother and Child Cohort Study (MoBa) is supported by the Norwegian Ministry of Health and Care Services and the Ministry of Education and Research, NIH/NIEHS (contract no N01-ES-75558), and NIH/NINDS (grant no. 1: UO1 NS 047537-01 and grant no. 2: UO1 NS 047537-06A1). The genotyping and analyses were supported by grants from Jane and Dan Olsson Foundation (Gothenburg, Sweden), Swedish Medical Research Council (2015-02559), Norwegian Research Council/FUGE (grant no. 151918/S10; FRI-MEDBIO 249779), March of Dimes (21-FY16-121), and the Burroughs Wellcome Fund Preterm Birth Research Grant (10172896) and by Swedish government grants to researchers in the public health sector (ALFGBG-717501, ALFGBG-507701, ALFGBG-426411) to BJ. RMF is supported by a Sir Henry Dale Fellowship (Wellcome Trust and Royal Society grant: WT104150). BF was supported by a grant from the Oak Foundation. DAL and GDS work in a unit that is supported by the University of Bristol and Medical Research Council (MC_UU_00011/1 and MC_UU_00011/6). DAL is supported by a grant from the US National Institute of Health (R01 DK10324), an NIHR Senior Investigator Award (NF-0616-10102), a grant from the European Research Council (DevelopObese; 669545) and a grant from the British Heart Foundation (AA/18/7/34219). The UK Medical Research Council and Wellcome (grant ref: 102215/2/13/2) and the University of Bristol provide core support for ALSPAC. GWAS data was generated by Sample Logistics and Genotyping Facilities at Wellcome Sanger Institute and LabCorp (Laboratory Corporation of America) using support from 23andMe. The DNBC data sets used for the analyses described in this manuscript were obtained from dbGaP at https://www.ncbi.nlm.nih.gov/gap/ through dbGaP accession number phs000103.v1.p1. The GWAS of Prematurity and its Complications study is one of the genome-wide association studies funded as part of the Gene Environment Association Studies (GENEVA) under the Genes, Environment and Health Initiative (GEI). The HAPO data sets used for the analyses described in this manuscript were obtained from dbGaP at https://www.ncbi.nlm.nih.gov/gap/ through dbGaP accession number phs000096.v4.p1. This study is part of the Gene Environment Association Studies initiative (GENEVA) funded by the trans-NIH Genes, Environment, and Health Initiative (GEI). The GPN datasets used for the analyses described in this manuscript were

## Author summary

### Why was this study done?

- Maternal height, BMI, blood glucose, and blood pressure are associated with gestational duration, birth weight, and birth length. These birth outcomes are subsequently associated with late-onset health conditions.

- The causal mechanisms and the relative contributions of maternal and fetal genetic effects underlying these observed associations are not clear.

### What did the researchers do and find?

- We dissected the relative contributions of maternal and fetal genetic effects using haplotype genetic score analysis in 10,734 mother–infant pairs of European ancestry.

- Genetically elevated maternal height is associated with longer gestational duration and larger birth size. In the fetus, alleles associated with adult height are positively associated with birth size.

- Alleles elevating blood pressure are associated with shorter gestational duration through a maternal effect and are associated with reduced fetal growth through a fetal genetic effect. Alleles that increase blood glucose in the mother are associated with increased birth weight, whereas risk alleles for type 2 diabetes in the fetus are associated with reduced birth weight.

- Alleles raising birth weight in fetus are associated with shorter gestational duration and higher maternal blood pressure during pregnancy.

### What do these findings mean?

- Maternal size and fetal growth are important factors in shaping the duration of gestation.

- Fetal growth is influenced by both maternal and fetal effects. Higher maternal BMI and glucose levels positively associate with birth weight through maternal effects. In the fetus, alleles associated with higher metabolic risks are negatively associated with birth weight.

- More rapid fetal growth is associated with shorter gestational duration and higher maternal blood pressure.

- These maternal and fetal genetic effects can largely explain the observed associations between maternal phenotypes and birth outcomes, as well as the life-course associations between these birth outcomes and adult phenotypes.

obtained from dbGaP at https://www.ncbi.nlm.nih.gov/gap/ through dbGaP accession number phs000714.v1.p1. Samples and associated were provided by the NICHD-funded Genomic and Proteomic Network for Preterm Birth Research (GPN-PBR). The funders had no role in study design, data collection and analysis, decision to publish, or preparation of the manuscript.

**Competing interests:** I have read the journal's policy and the authors of this manuscript have the following competing interests: DAL has received support, via her University, from Roche Diagnostics and Medtronic Ltd for biomarker research unrelated to this study. LJM consults for Mirvie, Inc., a biotech company involved in preterm birth diagnostics. The content of this paper is unrelated to the areas of their diagnostics development. GDS is a member of the Editorial Board of PLOS Medicine. The other authors report no conflicts.

**Abbreviations:** ALSPAC, The Avon Longitudinal Study of Parents and Children; BMI, body mass index; BP, blood pressure; DBP, diastolic blood pressure; DNBC, The Danish National Birth Cohort; FIN, The Finnish birth data set; FPG, fasting plasma glucose; GPN, The Genomic and Proteomic Network for Preterm Birth Research; GWA, genome-wide association; HAPO, Hyperglycemia and Adverse Pregnancy Outcome; MoBa, The Mother Child data set of Norway; MR, mendelian randomization; MR-PRESSO, MR pleiotropy residual sum and outlier; OGTT, oral glucose tolerance test; PCA, principal components analysis; SBP, systolic blood pressure; SD, standard deviation; SNP, single-nucleotide polymorphism; STROBE, Strengthening the Reporting of Observational Studies in Epidemiology; TSLS, two-stage least squares; T2D, type 2 diabetes.

## Introduction

Epidemiological studies have demonstrated that maternal physical and physiological traits associate with birth outcomes. For example, maternal height is positively associated with gestational duration [1, 2], birth weight, and birth length [3, 4]; higher maternal blood glucose is associated with higher birth weight [5]; and elevated maternal blood pressure (BP) is associated with reduced birth weight [6, 7]. These birth outcomes in turn associate with many long-term adverse health outcomes in the offspring, such as obesity [8], type 2 diabetes (T2D) [9], hypertension [10], and cardiovascular diseases [11, 12]. Different mechanisms have been proposed to explain the observed associations between maternal phenotypes and pregnancy outcomes [13–16] (Fig 1A), as well as the life-course associations between birth outcomes and adult phenotypes [17–20] (Fig 1B). Briefly, these include various causal effects (for example, maternal effects, defined as the causal influence of the maternal phenotype on birth outcomes or offspring phenotype [21]), genetically confounded associations due to genetic sharing (between mothers and infants) or shared genetic effects (between a birth outcome and an adult phenotype) [20], and confounding due to the environment. Fetal phenotypes can also affect maternal physiology during or even after pregnancy (fetal drive) [22]. Dissecting these different underlying mechanisms would increase knowledge of the etiology of these critical pregnancy outcomes and provide insights into how pregnancy outcomes are linked with later-onset disorders [13, 23]. Understanding the causal effects of modifiable maternal phenotypes could have implications for clinical interventions to prevent adverse birth outcomes [24]. The shared genetic causes between pregnancy characteristics and late offspring outcomes could provide insights into the molecular pathways through which these shared genetic effects are mediated [20].

Mendelian randomization (MR) [25] studies utilizing maternal genotypes as instrumental variables have been used to probe the causal relationships between maternal phenotypes and pregnancy outcomes [13, 16, 26]. Using this approach, Tyrrell and colleagues [24] demonstrated that higher maternal body mass index (BMI) and blood glucose levels are causally associated with higher birth weight, whereas higher maternal systolic BP (SBP) causes lower birth weight. Using a genome-wide association (GWA) approach, Horikoshi and colleagues [20] demonstrated strong inverse genetic correlations between birth weight and adult cardiometabolic diseases, suggesting a strong genetic component underlying the observed associations between low birth weight and cardiometabolic risks. More recently, Warrington and colleagues estimated maternal and fetal genetic effects on birth weight genome-wide and investigated associations between those genetic effects on birth weight and adult SBP [27].

We previously developed an MR method that utilizes nontransmitted maternal alleles as a valid genetic instrument for maternal phenotypic effects on fetal/offspring outcomes [15]. We showed that the observed association between maternal height and fetal size is mainly due to shared genetics, whereas the association between maternal height and gestational duration is more likely causal. Studies based on this approach have provided novel understandings about the causal relationships between many maternal phenotypes and birth outcomes. They have also highlighted genetic contributions to life-course associations between birth weight and late-onset diseases [20, 27]. However, previous studies have usually examined causal effects of maternal phenotypes on either birth weight or gestational duration separately despite the strong association between them [28, 29]. The studies focusing on birth weight have not explored whether any effects on birth weight are driven by effects on gestational duration because the information was not always available. In addition, the causal effects of fetal growth on gestational duration and maternal phenotypes during pregnancy have not been previously investigated using genetic approaches.

To further our understanding of how various maternal phenotypes are correlated with pregnancy outcomes through maternal or fetal genetic effects and how fetal growth influences

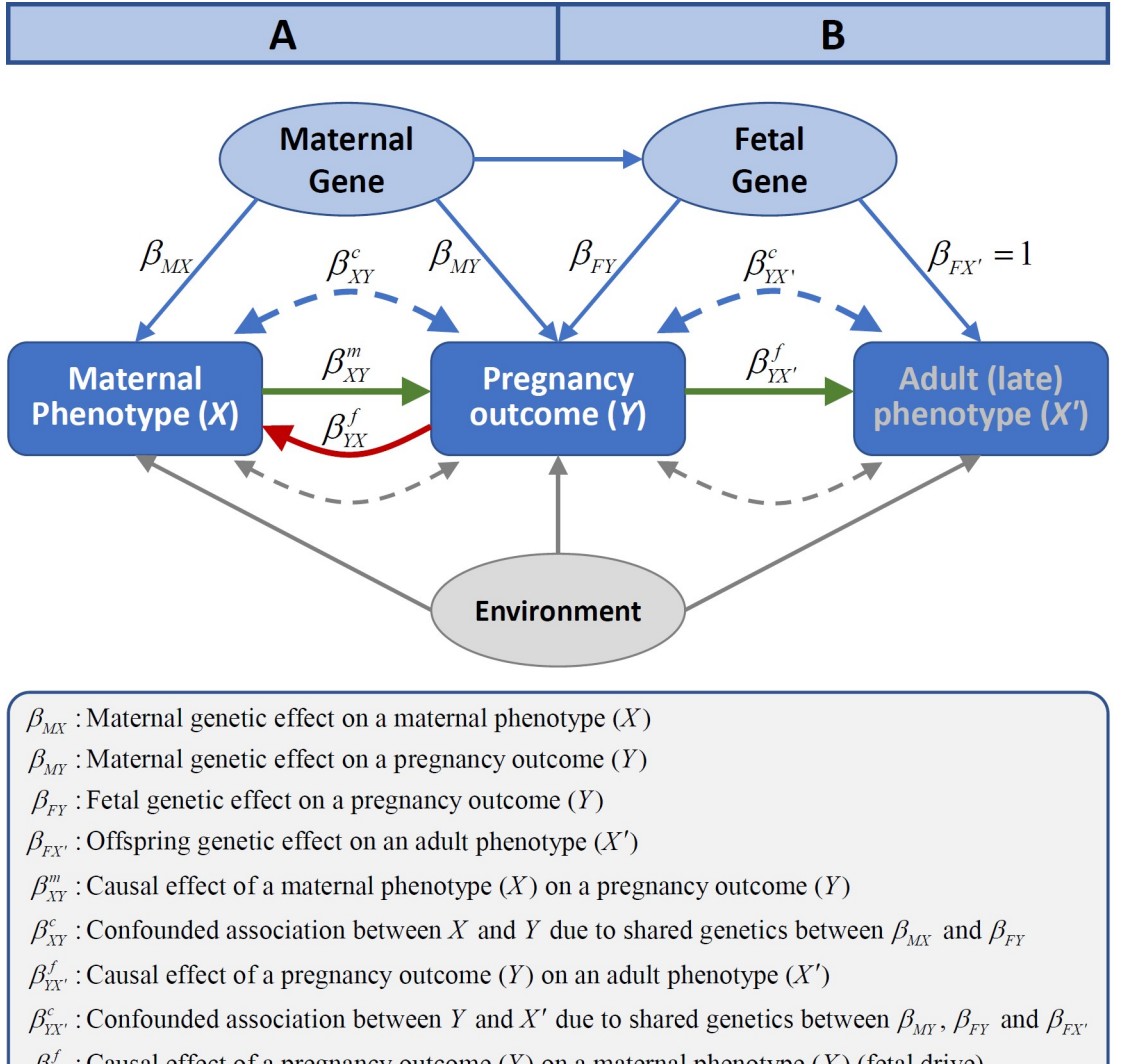

$\beta_{MX}$ : Maternal genetic effect on a maternal phenotype $(X)$

$\beta_{MY}$ : Maternal genetic effect on a pregnancy outcome $(Y)$

$\beta_{FY}$ : Fetal genetic effect on a pregnancy outcome $(Y)$

$\beta_{FX'}$ : Offspring genetic effect on an adult phenotype $(X')$

$\beta^m_{XY}$ : Causal effect of a maternal phenotype $(X)$ on a pregnancy outcome $(Y)$

$\beta^c_{XY}$ : Confounded association between $X$ and $Y$ due to shared genetics between $\beta_{MX}$ and $\beta_{FY}$

$\beta^f_{YX'}$ : Causal effect of a pregnancy outcome $(Y)$ on an adult phenotype $(X')$

$\beta^c_{YX'}$ : Confounded association between $Y$ and $X'$ due to shared genetics between $\beta_{MY}$, $\beta_{FY}$ and $\beta_{FX'}$

$\beta^f_{YX}$ : Causal effect of a pregnancy outcome $(Y)$ on a maternal phenotype $(X)$ (fetal drive)

**Fig 1. The different mechanisms underlying (A) the associations between maternal phenotypes and pregnancy outcomes and (B) the associations between pregnancy outcomes and late adult phenotypes in offspring.** These mechanisms include 1) causal effects of maternal phenotypes on pregnancy outcomes ($\beta^m_{XY}$) and causal effects of pregnancy outcomes on adult phenotypes ($\beta^f_{YX'}$) (green arrows), 2) genetically confounded associations between maternal phenotypes and pregnancy outcomes ($\beta^c_{XY}$) because of genetic sharing between mothers and infants and genetically confounded associations between birth outcomes and adult phenotypes in offspring ($\beta^c_{YX'}$) because of shared genetic effects (blue dashed arrows), 3) confounding due to environmental effects (gray dashed arrows, which were not examined in this study), and 4) fetal drive ($\beta^f_{YX}$)—fetus causally influencing maternal phenotypes during pregnancy (red arrow).

gestational duration and maternal physiological changes during pregnancy, we expanded our haplotype-based method by considering the mother–fetus duo (pregnancy) as the analytical unit [30] and explicitly modeled maternal and fetal genetic effects using haplotype genetic scores (Fig 2). By testing associations between these haplotype genetic scores and birth outcomes, we systematically investigated the maternal and fetal genetic effects underlying the observed associations between 4 maternal traits (height, BMI, BP, and blood glucose levels) and pregnancy outcomes (gestational duration, birth weight, and birth length). Using this approach, we also examined the associations between fetal growth (using gestational-age–

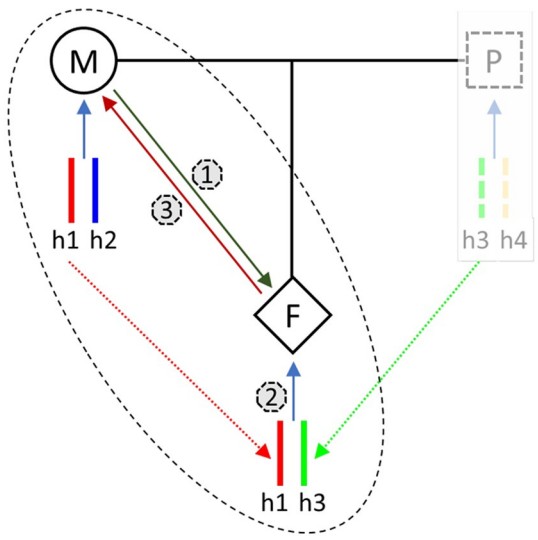

Association between a pregnancy phenotype $Y$ and the three haplotype genetic scores

$$Y = \alpha + \beta_{h1}S_{h1} + \beta_{h2}S_{h2} + \beta_{h3}S_{h3} + \mathbf{V}\boldsymbol{\beta}_{\mathbf{V}} + e$$

Effects of the three haplotype genetic scores and their interpretations

$\beta_{h1}$ : effect of maternal transmitted alleles (maternal and fetal genetic effect)

$\beta_{h2}$ : effect of maternal non-transmitted alleles (only maternal effect)

$\beta_{h3}$ : effect of paternal transmitted alleles (only fetal genetic effect)

Modeling maternal and fetal effects using linear combinations of the haplotype effects

$\beta_{MY} = (\beta_{h1} - \beta_{h3} + \beta_{h2}) / 2$ : maternal effect on $Y$ by the maternal alleles

$\beta_{FY} = (\beta_{h1} - \beta_{h2} + \beta_{h3}) / 2$ : fetal genetic effect on $Y$ by the fetal alleles

**Fig 2. Genetic dissection of maternal and fetal genetic effects using haplotype genetic scores in mother–child pairs.** There are 3 groups of alleles (haplotypes) in a mother (M)–fetus (F) duo: the maternal transmitted alleles (h1) can affect a pregnancy outcome through a maternal (1) and/or fetal genetic effect (2), the maternal nontransmitted alleles (h2) can only affect a pregnancy outcome through maternal effect (1), and the paternal transmitted alleles (h3) only through fetal effect (2) (assuming no paternal effect). The paternal transmitted alleles (h3) could influence a maternal phenotype during pregnancy by fetal drive (3). The paternal genetics might be able to influence maternal phenotype and pregnancy outcomes through the environment that the fathers create (i.e., paternal effect or "genetic nurture"). However, for the traits and their associated variants considered in this study (S2 Table), the paternal effects should be minimal and therefore were assumed to be zero.

adjusted birth weight as a measure of this) and pregnancy outcomes, maternal BP, and maternal blood glucose levels measured during pregnancy.

## Methods

A prospective protocol for analysis was not prepared for this study; however, the assembly of the data sets and all the analyses were planned in advance of data analysis. We reported this study according to the Strengthening the Reporting of Observational Studies in Epidemiology (STROBE) guideline [31] for cross-sectional studies (S1 STROBE Checklist). All data derived from the present study are presented with or in the paper.

### Data sets

We used phenotype and genome-wide single-nucleotide polymorphism (SNP) data of 10,734 mother–infant pairs from 6 birth studies (S1 Table). These include 3 case/control data sets

collected from Nordic countries (The Finnish birth data set [FIN], The Mother Child data set of Norway [MoBa], and The Danish National Birth Cohort [DNBC]) for genetic studies of preterm birth [32], a longitudinal birth cohort (the Avon Longitudinal Study of Parents and Children [ALSPAC]) [33] from the UK, a study of preterm birth from the US (The Genomic and Proteomic Network for Preterm Birth Research [GPN]) [34], and the Hyperglycemia and Adverse Pregnancy Outcome [HAPO] study [5] with samples of European ancestry collected from the UK, Canada, and Australia. A detailed description of these data sets can be found in the Supplementary Methods (S1 Text).

We focused on investigating the relationships between maternal height, prepregnancy BMI, BP, blood glucose levels, and pregnancy outcomes including gestational duration (as both quantitative and dichotomous preterm/term trait), birth weight, and birth length. Maternal height, prepregnancy BMI, and the 3 pregnancy outcomes were available in most of the studies (birth weight and length were not available in the MoBa data used here, and birth length was not available in the DNBC data used here) (Table 1). Maternal BP during pregnancy was available in ALSPAC and HAPO. In the HAPO data, BP was measured between 24 and 32 weeks of pregnancy when the mothers underwent an oral glucose tolerance test (OGTT) [5]. In ALSPAC, all BP measurements undertaken during antenatal care were extracted from clinical records; women had a median of 13 (interquartile range 11–16) BP measurements [35]. We used the average of the BPs measured between 30 to 36 weeks of gestation (as close as possible to when the BP was measured in HAPO). Maternal fasting plasma glucose (FPG) levels during pregnancy were available only in the HAPO study. FPG was measured in over 4,000 ALSPAC mothers in a follow-up data collection 18 years after the pregnancy (S1 Text and Table 1).

Because gestational duration is a key determinant of birth weight and birth size, we only included pregnancies with spontaneous deliveries and excluded mother–child pairs without gestational duration information. Pregnancies with known gestational or fetal complications and pre-existing medical conditions were excluded. Detailed inclusion/exclusion criteria are provided in Supplementary Methods (S1 Text). Preterm birth was defined as birth before 37 completed weeks of pregnancy.

This study involves reanalysis of existing data sets, and the proposed analytical aims are consistent with the original consent agreements under which the genomic and phenotypic data were obtained. Therefore, additional ethics approval was not required.

## Genotype data

Genome-wide SNP data were generated using either Affymetrix 6.0 (Affymetrix, Santa Clara, CA, USA) or various Illumina genotyping arrays (Illumina, San Diego, CA, USA). Standardized genotype quality control procedures were applied to all data sets. Participants of non-European ancestry were identified and excluded using principal components analysis (PCA) (S1 Text and S1 Fig). Genome-wide imputation was performed using Minimac3 [36] and the reference haplotypes from phase 3 of the 1000 Genomes Project [37]. In each data set, the haplotype phasing was done in all maternal and fetal samples using Shapeit2 [38]. This program accommodates mother–child relationship and accurately estimates mother–child allele transmission when phasing mother–child duos together.

## Construction of genetic scores

We constructed weighted genetic scores to instrument various maternal phenotypes using GWA SNPs and their estimated effect sizes reported by the most recent large GWA studies (S1 Text and S2 Table). Specifically, 2,130 height-associated SNPs and 628 BMI-associated SNPs reported by the GIANT consortium [39] were used to build genetic scores for height and BMI,

**Table 1. Descriptive statistics of maternal traits and birth outcomes.**

| Trait | FIN N | FIN Mean | FIN SD | MoBa N | MoBa Mean | MoBa SD | DNBC N | DNBC Mean | DNBC SD | HAPO N | HAPO Mean | HAPO SD | GPN N | GPN Mean | GPN SD | ALSPAC N | ALSPAC Mean | ALSPAC SD | Total N | Total Mean | Total SD |
|---|---|---|---|---|---|---|---|---|---|---|---|---|---|---|---|---|---|---|---|---|---|
| *Maternal traits* | | | | | | | | | | | | | | | | | | | | | |
| Height (cm) | 1,170 | 166.7 | 5.8 | 975 | 167.9 | 5.8 | 1,653 | 168.9 | 6.1 | 1,089 | 164.5 | 6.2 | 342 | 164.9 | 7.8 | 4,993 | 164.4 | 6.6 | 10,222 | 165.7 | 6.4 |
| BMI (kg/m²) | 1,168 | 23.0 | 3.7 | 958 | 24.0 | 4.2 | 1,625 | 23.5 | 4.2 | 1,051 | 24.2 | 4.7 | 337 | 24.2 | 5.3 | 4,759 | 22.9 | 3.7 | 9,898 | 23.3 | 4.0 |
| SBP (mmHg) | | NA | | | NA | | | NA | | 1,089 | 108.0 | 9.7 | | NA | | 4,923 | 113.1 | 6.8 | 6,012 | 112.1 | 7.4 |
| DBP (mmHg) | | | | | | | | | | 1,089 | 71.1 | 7.8 | | | | 4,943 | 66.8 | 4.7 | 6,032 | 67.6 | 5.4 |
| FPG (mmol/L)[a] | | | | | | | | | | 1,088 | 4.5 | 0.4 | | | | 2,452 | 5.2 | 0.4 | 3,540 | 5.0 | 0.4 |
| *Birth outcomes*[b] | | | | | | | | | | | | | | | | | | | | | |
| Gestational days | 892 | 282.0 | 6.3 | 522 | 280.4 | 3.6 | 977 | 282.6 | 3.0 | 1,034 | 281.2 | 7.2 | 197 | 276.4 | 4.2 | 4,967 | 279.8 | 7.9 | 8,589 | 280.5 | 7.0 |
| | 255 | 237.9 | 13.4 | 487 | 248.2 | 11.4 | 748 | 240.1 | 15.2 | | None | | 146 | 212.0 | 20.3 | 212 | 242.7 | 13.5 | 1,848 | 240.0 | 14.3 |
| Birth weight (g) | 892 | 3,578.0 | 434.3 | | NA | | 973 | 3,712.6 | 456.5 | 1,034 | 3,463.3 | 487.0 | 197 | 3,505.2 | 366.0 | 4,909 | 3,514.3 | 455.7 | 8,005 | 3,538.7 | 455.5 |
| | 255 | 2,348.8 | 477.1 | | | | 739 | 2,459.5 | 645.8 | | None | | 146 | 1,641.4 | 522.1 | 208 | 2,543.1 | 536.8 | 1,348 | 2,362.9 | 587.4 |
| Birth length (cm) | 892 | 50.4 | 1.9 | | NA | | | NA | | 1,034 | 50.7 | 2.2 | 192 | 34.7 | 1.4 | 4,004 | 51.0 | 2.1 | 6,122 | 50.4 | 2.1 |
| | 253 | 45.0 | 2.9 | | | | | | | | None | | 146 | 28.4 | 3.5 | 152 | 47.3 | 2.5 | 551 | 41.3 | 3.0 |
| **Male/Female**[c] | 606/564 | | | 503/506 | | | 908/831 | | | 550/539 | | | 189/154 | | | 2,690/2,694 | | | 5,446/5,288 | | |

[a]In HAPO, maternal FPG was measured between 24–32 weeks of gestation. In ALSPAC, the FPG was measured 18 years after pregnancy.

[b]Descriptive statistics of pregnancy outcomes were calculated in term (≥38 weeks, upper row) and preterm infants (<37 weeks, lower row) separately. There were no preterm infants in the HAPO data set.

[c]Number of male and female infants.

**Abbreviations:** ALSPAC, The Avon Longitudinal Study of Parents and Children; BMI, body mass index; DBP, diastolic blood pressure; DNBC, The Danish National Birth Cohort; FIN, The Finnish birth data set; FPG, fasting plasma glucose; GPN, The Genomic and Proteomic Network for Preterm Birth Research; HAPO, The Hyperglycemia and Adverse Pregnancy Outcome Study; MoBa, The Mother Child data set of Norway; NA, not available; SBP, systolic blood pressure; SD, standard deviation.

respectively. Eight hundred thirty-one SNPs associated with BP [40] were used to build genetic scores for BP. For simplicity, we built a score for average BP using the mean estimated effects of SBP and diastolic BP (DBP). For FPG, we used 22 SNPs associated with FPG levels identified in individuals without diabetes [41]. We also constructed a T2D genetic score using 306 T2D SNPs [42] (excluding SNPs overlapping or in close linkage disequilibrium with the 22 FPG SNPs) (S1 Text and S2 Table). To examine the associations of fetal growth (as proxied by birth weight) with pregnancy outcomes and maternal BP and FPG during pregnancy, we constructed genetic scores using 86 SNPs associated with birth weight with confirmed fetal effect [27]. The lists of these GWA SNPs used in constructing genetic scores are provided in S1 Data.

For each set of GWA SNPs, we constructed 2 genotype genetic scores: $S_{mat}$ (maternal genotype score), $S_{fet}$ (fetal genotype score), and 3 haplotype genetic scores: $S_{h1}$, $S_{h2}$, and $S_{h3}$, respectively, based on the maternal transmitted (h1), maternal nontransmitted (h2), and paternal transmitted alleles (h3) (Fig 2). It follows that $S_{mat} = S_{h1} + S_{h2}$ and $S_{fet} = S_{h1} + S_{h3}$.

## Statistical analyses

**Phenotype associations and instrumental strength of genetic scores.**   We first assessed the associations between the 4 maternal phenotypes ($X$) (i.e., height, BMI, BP, and FPG) and each pregnancy outcome ($Y$) (i.e., gestational duration, preterm birth, birth weight, and birth length) using regression analyses. Maternal age, fetal sex, maternal height, and prepregnancy BMI were included as covariates. Because gestational duration influences birth weight and length in a nonlinear fashion, the first 3 orthogonal polynomials of gestational duration were included as covariates in the analysis of birth weight and length. These analytical models are described in more detail in the Supplementary Methods (S1 Text).

The instrumental strength of the genetic scores was checked by the variance in a maternal phenotype explained ($R^2$) by the corresponding genetic scores.

**Association tests between haplotype genetic scores and pregnancy outcomes.**   Associations between the haplotype genetic scores and the pregnancy outcomes were tested using regression models like those used in the association analysis described above, except that the maternal phenotypes ($X$) were replaced by their corresponding 3 haplotype genetic scores ($S_{h1} + S_{h2} + S_{h3}$). The associations between these haplotype scores can differentiate between maternal and fetal genetic effects (Fig 2). Specifically, an association of $S_{h2}$ (maternal nontransmitted haplotype score) with a pregnancy outcome suggests a maternal (intrauterine phenotypic) effect, whereas an association of $S_{h3}$ (paternal transmitted haplotype score) with the pregnancy outcomes suggests fetal genetic effects. The 3 haplotype genetic scores of the same mother–child pairs ($S_{h1}$, $S_{h2}$, and $S_{h3}$) were simultaneously tested in a single-regression model (i.e., $Y$ is modeled as a function of $S_{h1} + S_{h2} + S_{h3} + Cov$, where Cov is a list of appropriate covariates), and hence, they had exactly the same sample size. Therefore, the effect size estimates of these haplotype scores and their associated $p$-values can be directly compared to assess the directions and relative contributions of the maternal and fetal effects.

**Modeling of maternal and fetal genetic effects.**   Whereas $S_{h2}$ and $S_{h3}$ can be used to draw inference about maternal and fetal genetic effects, this question can be addressed with greater statistical power by also including $S_{h1}$, the maternal transmitted haplotype score, in the model. Thus, we modeled the maternal effect and fetal genetic effect as different linear combinations of the regression coefficients of the 3 haplotype genetic scores (Fig 2 and S1 Text) [30]. Under the assumptions of additivity between maternal and fetal effect and zero parent-of-origin effect, the total effect ($\beta_{h1}$) of the maternal transmitted haplotype (h1) should be equal to the summation of the maternal effect ($\beta_{h2}$) of the nontransmitted haplotype (h2) and fetal genetic effect ($\beta_{h3}$) of the paternal transmitted haplotype (h3). Thus, ($\beta_{h1}-\beta_{h3}$) and ($\beta_{h1}-\beta_{h2}$),

respectively, represent the maternal effect and the fetal genetic effect of the maternal transmitted haplotype (h1). Therefore, the average maternal effect ($\beta_{MY}$) of the 2 maternal haplotypes on a birth outcome ($Y$) can be expressed as ($\beta_{h1}-\beta_{h3}+\beta_{h2}$)/2, and the average fetal genetic effect ($\beta_{FY}$) of the maternal and paternal transmitted haplotypes can be expressed as ($\beta_{h1}-\beta_{h2}+\beta_{h3}$)/2. Because these linear combinations also capture the maternal or fetal genetic effects of the maternal transmitted haplotype (h1), they are more powerful than the methods only using the maternal nontransmitted haplotype (h2) or the paternal transmitted haplotype (h3) as instruments, respectively, for maternal effect and fetal genetic effect.

**Estimation of maternal causal effects.** The estimated maternal effect ($\hat{\beta}_{MY}$) from the haplotype genetic score association analyses can be interpreted as the amount of change in a pregnancy outcome ($Y$) caused by a certain amount of difference in a maternal phenotype ($X$) associated with one-unit genetic score. The maternal causal effect ($\hat{\beta}_{XY}^{m}$) was estimated using the ratio estimate [43] ($\hat{\beta}_{MY}/\hat{\beta}_{MX}$), where $\hat{\beta}_{MX}$ is the estimated maternal effect on the maternal phenotype (Fig 2). As an alternative, we also performed instrumental variable analysis using the two-stage least-squares (TSLS) approach [43], with the maternal nontransmitted haplotype score ($S_{h2}$) as the genetic instrument for maternal causal effect [15].

**Estimation of genetically confounded associations.** The fetal genetic effect ($\beta_{FY}$) reflects that the genetic variants associated with an adult phenotype ($X'$) in the offspring or the corresponding maternal phenotype ($X$) have direct fetal genetic effect on a pregnancy outcome ($Y$). This shared genetic effect can confound the association between a maternal phenotype ($X$) and a pregnancy outcome ($Y$), as well as the association between the pregnancy outcome ($Y$) and the adult phenotype ($X'$) in offspring (Fig 1).

By assuming that all the genetic variants associated with an adult phenotype in offspring ($X'$) or the corresponding maternal phenotype ($X$) have a similar effect on a pregnancy outcome ($Y$) as the fetal genetic effect estimated from the genetic score built on known GWA SNPs ($\hat{\beta}_{FY}$), we can approximately estimate the magnitude of these genetically confounded associations (see S1 Text for details). Specifically, the genetically confounded association between a maternal phenotype ($X$) and a pregnancy outcome ($Y$) due to the shared genetic effect can be estimated by

$$\hat{\beta}_{XY}^{c} = \frac{\frac{h_X^2}{2}\hat{\beta}_{FY}}{\hat{\beta}_{MX}},$$

where $\hat{\beta}_{MX}$ is estimated maternal effect and $h_X^2$ is the heritability (the proportion of additive genetic variance) of the maternal phenotype ($X$).

Similarly, the genetically confounded association between a pregnancy outcome ($Y$) and an adult (late) phenotype ($X'$) in offspring can be estimated by

$$\hat{\beta}_{YX'}^{c} = h^2 \frac{\text{Var}(X')}{\text{Var}(Y)} \left( \frac{\hat{\beta}_{MY}}{2 + \hat{\beta}_{FY}} \right) \qquad \text{(Method 1)}$$

and

$$\hat{\beta}_{YX'}^{c} = h^2 \frac{\text{Var}(X')}{\text{Var}(Y)} \left( \frac{\widehat{\beta_{h1} + \beta_{h3}}}{2} \right), \qquad \text{(Method 2)}$$

where $h^2$ is the heritability of the adult phenotype and $\text{Var}(X')$ and $\text{Var}(Y)$ are, respectively, the variance of the adult phenotype and the variance of the pregnancy outcome. The first method

(Method 1) can partition the confounded association into the maternal ($\frac{\beta_{MY}}{2}$) and the fetal component ($\beta_{FY}$) (S1 Text).

**Multivariable MR analysis.** The genetic scores built on hundreds or thousands of SNPs are likely to be less specific because they are more likely to be associated with other phenotypic traits [44], which can introduce ambiguities in the interpretation of genetic score associations [45]. To circumvent this issue, we performed a two-sample multivariable MR analysis [46, 47] using the MR pleiotropy residual sum and outlier (MR-PRESSO) [48] to detect and correct for variants with horizontal pleiotropic effects [49] in multiple-variant MR testing. Three MR-PRESSO tests were applied: the global test was used to detect the presence of horizontal pleiotropy, the outlier test to identify variants with significant horizontal pleiotropic effect, and the distortion test to estimate the distortion caused by significant horizontal pleiotropic outlier variants. This analysis studies the maternal or fetal genetic effects by testing whether the effects of the maternal transmitted (h1), maternal nontransmitted (h2), and paternal transmitted (h3) alleles of the GWA SNPs on a pregnancy outcome are proportional to their reported effects on an adult phenotype ($X'$) in the reference GWA studies (S1 Text and S4 Fig). The allele-specific effect estimates (for the h1, h2, and h3 alleles) of each SNP on a pregnancy outcome were obtained using the same regression methods for haplotype genetic score analysis (S1 Text).

We did meta-analyses of the results from all available data sets to generate the overall results. Fixed-effect meta-analysis was used to combine the regression coefficients and standard errors from individual studies, and we checked between-study heterogeneity using Cochran's Q test. The meta-analyses were done using the R metafor package [50].

## Results

### Phenotypic associations between maternal phenotypes and pregnancy outcomes

We used 10,734 mother–infant pairs with both genotype and phenotype data in our analyses (S1 Table). Distributions of key variables for the maternal phenotypes and pregnancy outcomes are shown in Table 1 and S5–S7 Figs.

The meta-analysis across the 6 data sets showed that taller maternal height was associated with longer gestational duration (0.14 day/cm, 95% CI: 0.10 to 0.18, $p = 2.2 \times 10^{-12}$), lower preterm birth risk (OR = 0.97 /cm, 95% CI: 0.96 to 0.98, $p = 2.2 \times 10^{-9}$), and higher birth weight (15 g/cm, 95% CI: 13.7 to 16.3, $p = 1.5 \times 10^{-111}$) and length (0.068 cm/cm, 95% CI: 0.061 to 0.075, $p = 1.6 \times 10^{-75}$). Maternal BMI was positively associated with birth weight (15.6 g/[kg/m$^2$], 95% CI: 13.5 to 17.7, $p = 1.0 \times 10^{-47}$) and birth length (0.05 cm/[kg/m$^2$], 95% CI: 0.04 to 0.06, $p = 3.9 \times 10^{-15}$) but was not associated with gestational duration (0.05 day per kg/m$^2$, 95% CI: −0.01 to 0.11, $p = 0.12$) or preterm birth risk (OR = 0.99 per kg/m$^2$, 95% CI: 0.98 to 1.01, $p = 0.42$) (S3 Table).

Using data from ALSPAC and HAPO, we observed that maternal BP during pregnancy was negatively associated with gestational duration and birth weight. The estimated effect sizes on gestational duration by SBP and DBP were −0.04 day/mmHg (95% CI: −0.08 to −0.01, $p = 7.3 \times 10^{-3}$) and −0.11 day/mmHg (95% CI: −0.15 to −0.06, $p = 3.3 \times 10^{-6}$), respectively. The estimated effect sizes on birth weight by SBP and DBP were −3.0 g/mmHg (95% CI: −4.6 to −1.5, $p = 1.8 \times 10^{-4}$) and −6.2 g/mmHg (95% CI: −8.4 to −3.9, $p = 6.0 \times 10^{-8}$), respectively. In HAPO, there was a strong positive association between maternal FPG and birth weight (192 g/[mmol/L], 95% CI: 116 to 268, $p = 5.7 \times 10^{-7}$) and birth length (0.62 cm/[mmol/L], 95% CI: 0.27 to 0.97, $p = 4.8 \times 10^{-4}$). However, the association between FPG measured 18 years after pregnancy with either birth weight or length in the ALSPAC data set was close to zero with wide confidence intervals (S3 Table).

## Associations between genetic scores and maternal phenotypes

We examined the instrumental strength of the genetic scores for the various maternal phenotypes. The maternal genotype genetic scores ($S_{mat}$) were associated with the corresponding maternal phenotypes and explained a substantial fraction of the phenotypic variances with similar contributions from the transmitted (h1) or the nontransmitted haplotype scores (h2) (S4 Table).

The maternal height genotype score ($S_{mat}$) explained >20% of the maternal height variance (S5 Table), and the maternal BMI genotype score ($S_{mat}$) explained approximately 5% of the maternal BMI variance (S6 Table). The BP genotype scores explained over 2% variance in maternal BP (S7 Table), which is less than half of the reported fraction of variance explained by the same score (5.7%) in a published GWA study of nonpregnant women and men (generally of an older age than pregnant women) [40], suggesting that these BP SNPs have a larger effect on BP in older populations or a weaker effect on maternal BP during pregnancy.

In HAPO, the maternal FPG genetic score built on 22 SNPs explained 8.3% of the FPG variance measured between 24–32 weeks. In ALSPAC, the same score explained 4.1% of the variance of FPG measured 18 years after pregnancy. By contrast, the T2D score (306 SNPs) explained much less FPG variance (S8 Table).

For each phenotype, we checked the correlations among the various genotype and haplotype genetic scores (S9 Table). For height, we observed significant correlations between the maternal genotype ($S_{mat}$) and the paternal transmitted haplotype score ($S_{h3}$) and between the maternal transmitted ($S_{h1}$) and nontransmitted scores ($S_{h2}$), indicating assortative mating [15, 51, 52] and increased homozygosity of height-associated SNPs.

## Maternal causal effects and genetically confounded associations between maternal phenotypes and birth outcomes

We next utilized haplotype genetic scores as genetic instruments to dissect the maternal and fetal genetic effects underlying the observed associations between maternal phenotypes and pregnancy outcomes (Table 2). Detailed meta-analysis of individual data sets can be found in S8–S12 Figs. We also conducted random-effects meta-analyses (S10 Table), and the results were essentially the same as the results obtained by fixed-effect meta-analyses (Table 2). To further check the robustness of the results, we conducted the analyses separately in ALSPAC (S11 Table) and the other 5 data sets (S12 Table). The results were similar, except the 5 data sets showed more associations with preterm birth, probably because these data sets had more preterm pregnancies. Based on the estimated maternal and fetal genetic effects, we estimated the maternal causal effects and genetically confounded associations between maternal phenotypes and birth outcomes due to shared genetics (Methods and Fig 3).

**Maternal height.**   The maternal nontransmitted height genetic score ($S_{h2}$) was positively associated with gestational duration ($p = 2.2 \times 10^{-4}$) and negatively associated with preterm birth ($p = 9.7 \times 10^{-4}$) (Table 2). The ratio estimates showed a maternal causal effect of approximately 1.0 days (95% CI: 0.38 to 1.64, $p = 1.8 \times 10^{-3}$) longer gestation per 1-standard deviation (SD) (6.4 cm) increase in maternal height. This effect was offset by a weaker and opposite fetal genetic effect of 0.71 days' (95% CI: 0.07 to 1.35, $p = 2.9 \times 10^{-2}$) shorter gestation per the same amount of genetic score associated with a 1-SD increase in maternal height (Fig 3 and S13 Table).

Maternal and paternal transmitted haplotype scores ($S_{h1}$ and $S_{h3}$) for height were positively associated with birth weight and birth length. The maternal nontransmitted score ($S_{h2}$) was also positively associated with birth weight and length, but the effect estimates were smaller

**Table 2. Association between haplotype genetic scores and birth outcomes.**

| Maternal Trait (Unit) Haplotype Score Tests[a] | Gestational Days | | | Preterm Birth (log[OR]) | | | Birth Weight (g) | | | Birth Length (cm) | | |
|---|---|---|---|---|---|---|---|---|---|---|---|---|
| | Beta[c] | SE | p-Value | Beta | SE | p-Value | Beta | SE | p-Value | Beta | SE | p-Value |
| **Height (cm)** | | | | | | | | | | | | |
| Maternal transmitted ($\beta_{h1}$) | 0.038 | 0.046 | 0.41 | −0.027 | 0.011 | 0.016* | 20 | 1.6 | $6.10 \times 10^{-38*}$ | 0.097 | 0.0087 | $1.10 \times 10^{-28*}$ |
| Maternal nontransmitted ($\beta_{h2}$) | 0.17 | 0.047 | 0.00022* | −0.037 | 0.011 | 0.00097* | 6.3 | 1.6 | $7.60 \times 10^{-5*}$ | 0.022 | 0.0088 | 0.011* |
| Paternal transmitted ($\beta_{h3}$) | −0.041 | 0.046 | 0.37 | 0.023 | 0.011 | 0.038* | 14 | 1.6 | $3.10 \times 10^{-19*}$ | 0.074 | 0.0085 | $4.70 \times 10^{-18*}$ |
| Maternal effect ($\beta_{MY}$) | 0.13 | 0.04 | 0.0017* | −0.044 | 0.0098 | $9.00 \times 10^{-6*}$ | 6.3 | 1.4 | $5.20 \times 10^{-6*}$ | 0.022 | 0.0076 | 0.0033* |
| Fetal effect ($\beta_{FY}$) | −0.089 | 0.041 | 0.029* | 0.016 | 0.0099 | 0.1 | 14 | 1.4 | $1.10 \times 10^{-23*}$ | 0.074 | 0.0076 | $2.70 \times 10^{-22*}$ |
| **BMI (kg/m²)** | | | | | | | | | | | | |
| Maternal transmitted ($\beta_{h1}$) | −0.059 | 0.17 | 0.73 | −0.065 | 0.041 | 0.11 | 23 | 5.8 | $7.00 \times 10^{-5*}$ | 0.066 | 0.032 | 0.038* |
| Maternal nontransmitted ($\beta_{h2}$) | −0.11 | 0.17 | 0.52 | 0.015 | 0.041 | 0.71 | 8.8 | 5.8 | 0.13 | 0.079 | 0.032 | 0.013* |
| Paternal transmitted ($\beta_{h3}$) | 0.027 | 0.17 | 0.88 | 0.049 | 0.04 | 0.23 | −6.5 | 5.7 | 0.26 | 0.027 | 0.032 | 0.39 |
| Maternal effect ($\beta_{MY}$) | −0.098 | 0.15 | 0.51 | −0.048 | 0.036 | 0.18 | 19 | 5 | 0.00016* | 0.06 | 0.028 | 0.031* |
| Fetal effect ($\beta_{FY}$) | 0.043 | 0.15 | 0.77 | −0.018 | 0.036 | 0.6 | 4 | 5 | 0.43 | 0.0077 | 0.028 | 0.78 |
| **BP[b] (mmHg)** | | | | | | | | | | | | |
| Maternal transmitted ($\beta_{h1}$) | −0.22 | 0.064 | 0.00067* | 0.034 | 0.015 | 0.027* | −6.8 | 2.2 | 0.0016* | −0.018 | 0.012 | 0.13 |
| Maternal nontransmitted ($\beta_{h2}$) | −0.035 | 0.064 | 0.59 | 0.047 | 0.016 | 0.0023* | −3.1 | 2.2 | 0.16 | −0.018 | 0.012 | 0.13 |
| Paternal transmitted ($\beta_{h3}$) | −0.016 | 0.064 | 0.8 | 0.005 | 0.015 | 0.74 | −5.9 | 2.1 | 0.0053* | −0.017 | 0.012 | 0.15 |
| Maternal effect ($\beta_{MY}$) | −0.12 | 0.055 | 0.033* | 0.038 | 0.013 | 0.0045* | −2 | 1.9 | 0.27 | −0.0096 | 0.01 | 0.35 |
| Fetal effect ($\beta_{FY}$) | −0.1 | 0.056 | 0.075 | −0.0035 | 0.013 | 0.8 | −4.8 | 1.9 | 0.0094* | −0.0086 | 0.01 | 0.4 |
| **FPG (mmol/L)** | | | | | | | | | | | | |
| Maternal transmitted ($\beta_{h1}$) | −3.9 | 1.8 | 0.029* | 0.6 | 0.43 | 0.16 | 13 | 59 | 0.82 | −0.067 | 0.32 | 0.84 |
| Maternal nontransmitted ($\beta_{h2}$) | −3.2 | 1.8 | 0.071 | 0.54 | 0.43 | 0.21 | 270 | 59 | $4.70 \times 10^{-6*}$ | 0.71 | 0.32 | 0.029* |
| Paternal transmitted ($\beta_{h3}$) | 2.7 | 1.7 | 0.12 | −0.38 | 0.43 | 0.37 | −52 | 59 | 0.38 | −0.13 | 0.32 | 0.69 |
| Maternal effect ($\beta_{MY}$) | −5 | 1.5 | 0.0012* | 0.77 | 0.37 | 0.039* | 170 | 51 | 0.0011* | 0.37 | 0.28 | 0.19 |
| Fetal effect ($\beta_{FY}$) | 0.99 | 1.5 | 0.51 | −0.14 | 0.36 | 0.7 | −150 | 50 | 0.0022* | −0.46 | 0.28 | 0.096 |
| **T2D (log[OR])** | | | | | | | | | | | | |
| Maternal transmitted ($\beta_{h1}$) | 0.013 | 0.3 | 0.97 | −0.0079 | 0.074 | 0.91 | −14 | 10 | 0.17 | −0.065 | 0.056 | 0.25 |
| Maternal nontransmitted ($\beta_{h2}$) | 0.05 | 0.31 | 0.87 | 0.013 | 0.072 | 0.86 | 33 | 10 | 0.0012* | 0.059 | 0.057 | 0.3 |
| Paternal transmitted ($\beta_{h3}$) | 0.83 | 0.31 | 0.0069* | −0.11 | 0.073 | 0.13 | −28 | 10 | 0.0061* | −0.033 | 0.057 | 0.56 |
| Maternal effect ($\beta_{MY}$) | −0.39 | 0.27 | 0.15 | 0.058 | 0.063 | 0.36 | 24 | 8.9 | 0.0064* | 0.015 | 0.049 | 0.76 |
| Fetal effect ($\beta_{FY}$) | 0.4 | 0.27 | 0.14 | −0.067 | 0.062 | 0.28 | −39 | 8.8 | $1.30 \times 10^{-5*}$ | −0.085 | 0.049 | 0.082 |

[a]$\beta_{h1}$, $\beta_{h2}$, and $\beta_{h3}$ are the effects of the 3 haplotype genetic scores. $\beta_{MY} = (\beta_{h1} − \beta_{h3} + \beta_{h2})/2$ and $\beta_{FY} = (\beta_{h1} − \beta_{h2} + \beta_{h3})/2$ are, respectively, the maternal and fetal genetic effects modeled by linear combinations of the haplotype effects.

[b]BP: mean of the SBP and DBP scores.

[c]Beta: estimated effects of genetic score associations given by per unit change in genetic scores of the maternal traits.

*p-Values less than 0.05.

**Abbreviations:** BMI, body mass index; BP, blood pressure; DBP, diastolic BP; SBP, systolic BP; SE, standard error; T2D, type 2 diabetes.

than the transmitted haplotype scores (Table 2). The larger effects of transmitted alleles indicate height-associated SNPs can influence growth in early prenatal development through fetal genetic effect. The estimates for maternal causal and fetal genetic effects were 50 g (95% CI: 29 to 72, $p = 5.6 \times 10^{-6}$) and 111 g (95% CI: 89 to 133, $p = 5.0 \times 10^{-23}$), respectively, for birth weight and 0.18 cm (95% CI: 0.06 to 0.30, $p = 3.3 \times 10^{-3}$) and 0.59 cm (95% CI: 0.47 to 0.71, $p = 1.0 \times 10^{-21}$) for birth length per genetic alleles associated with a 1-SD (6.4 cm) increase in maternal height (Fig 3, S13 Fig, and S13 Table).

**Maternal prepregnancy BMI.** BMI haplotype genetic scores were not significantly associated with gestational duration or preterm birth (Table 2), and the ratio estimates of maternal causal effects on gestational duration (−0.45 day, 95% CI: −1.80 to 0.89, $p = 0.51$) and preterm birth (OR = 0.8, 95% CI: 0.6 to 1.1, $p = 0.18$) were not significant (S13 Table), suggesting both minimal maternal and fetal effect of the BMI-associated SNPs on gestational duration. Linear hypotheses modeling suggested that BMI-associated SNPs have some maternal and no fetal effect on birth weight and length (Table 2). The estimated maternal causal effect on birth weight was 88 g (95% CI: 42 to 134, $p = 1.9 \times 10^{-4}$) and on birth length was 0.28 cm (95% CI: 0.02 to 0.53, $p = 3.2 \times 10^{-2}$) per 1-SD (4.0 kg/m$^2$) genetically increased BMI (Fig 3, S13 Fig, and S13 Table).

**Maternal BP.** There was a significant association between the maternal nontransmitted BP genetic score and increased preterm birth risk ($p = 2.3 \times 10^{-3}$), suggesting the association between maternal BP and gestational duration was primarily driven by a maternal effect (Table 2). The estimated causal effect sizes based on the ratio method were −2.3 days (95% CI: −4.4 to −0.14, $p = 3.6 \times 10^{-2}$) on gestational duration and OR = 2.1 (95% CI: 1.2 to 3.5, $p = 5.7 \times 10^{-3}$) in preterm birth risk per 1-SD (5.8 mmHg) genetically increased maternal BP (Fig 3, S13 Fig, and S13 Table).

Both maternal transmitted and paternal transmitted BP scores were negatively associated with birth weight ($p = 1.6 \times 10^{-3}$ and $5.3 \times 10^{-3}$ for $S_{h1}$ and $S_{h3}$, respectively; Table 2), suggesting the negative association between maternal BP and birth weight was mainly caused by a fetal genetic effect. The estimated fetal genetic effect was a reduction of 94 g (95% CI: 21 to 166, $p = 1.1 \times 10^{-2}$) in birth weight by alleles associated with a 1-SD (5.8 mmHg) increase in maternal BP (Fig 3 and S13 Table).

**Maternal FPG.** We observed a positive association between maternal nontransmitted FPG genetic score ($S_{h2}$) and birth weight ($p = 4.7 \times 10^{-6}$), indicating a strong causal effect of increased maternal FPG level on higher birth weight (Table 2). In the HAPO data set, the estimated causal effect size by TSLS based on ($S_{h2}$) was 147 g (95% CI: 15 to 279, $p = 3.0 \times 10^{-2}$) per 1-SD (0.36 mmol/L) increase in maternal FPG, and the ratio estimate in all mother–child pairs was 59 g (95% CI: 22 to 96, $p = 1.8 \times 10^{-3}$) (S13 Table). Interestingly, but not unexpectedly, the linear modeling showed a negative fetal genetic effect of FPG increasing alleles on birth weight (−54 g, 95% CI: −90 to −18, $p = 3.3 \times 10^{-3}$). The linear hypothesis modeling also showed a significant negative maternal effect of high maternal FPG on gestational duration (−1.7 days, 95% CI: −2.8 to −0.6, $p = 2.0 \times 10^{-3}$) and increased risk for preterm birth (OR = 1.3, 95% CI: 1.0 to 1.7, $p = 4.3 \times 10^{-2}$) per 1-SD (0.36 mmol/L) increase in maternal FPG (Fig 3, S13 Fig, and S13 Table).

To further understand whether the observed associations were driven by the SNPs that influence normal FPG levels or by the SNPs associated with pathological hyperglycemia (i.e., T2D), we tested associations between the T2D haplotype scores and pregnancy outcomes. Compared to the FPG scores, the maternal nontransmitted T2D score ($S_{h2}$) was less significantly associated with both birth weight and gestational duration, but the negative associations between the paternal transmitted score ($S_{h3}$) and birth weight and gestational duration were more apparent (Table 2).

As shown in Fig 3 and S13 Fig, the combinations of the estimated maternal causal effects and the genetically confounded associations due to genetic transmission were consistent with the observed phenotypic associations between maternal phenotypes and birth outcomes. The maternal causal effects were usually more dominant than the genetically confounded associations in shaping the phenotypic associations between maternal phenotypes and birth outcomes. In some cases, the maternal causal effects and the genetically confounded associations pointed to opposite directions (for example, between height and gestational duration and

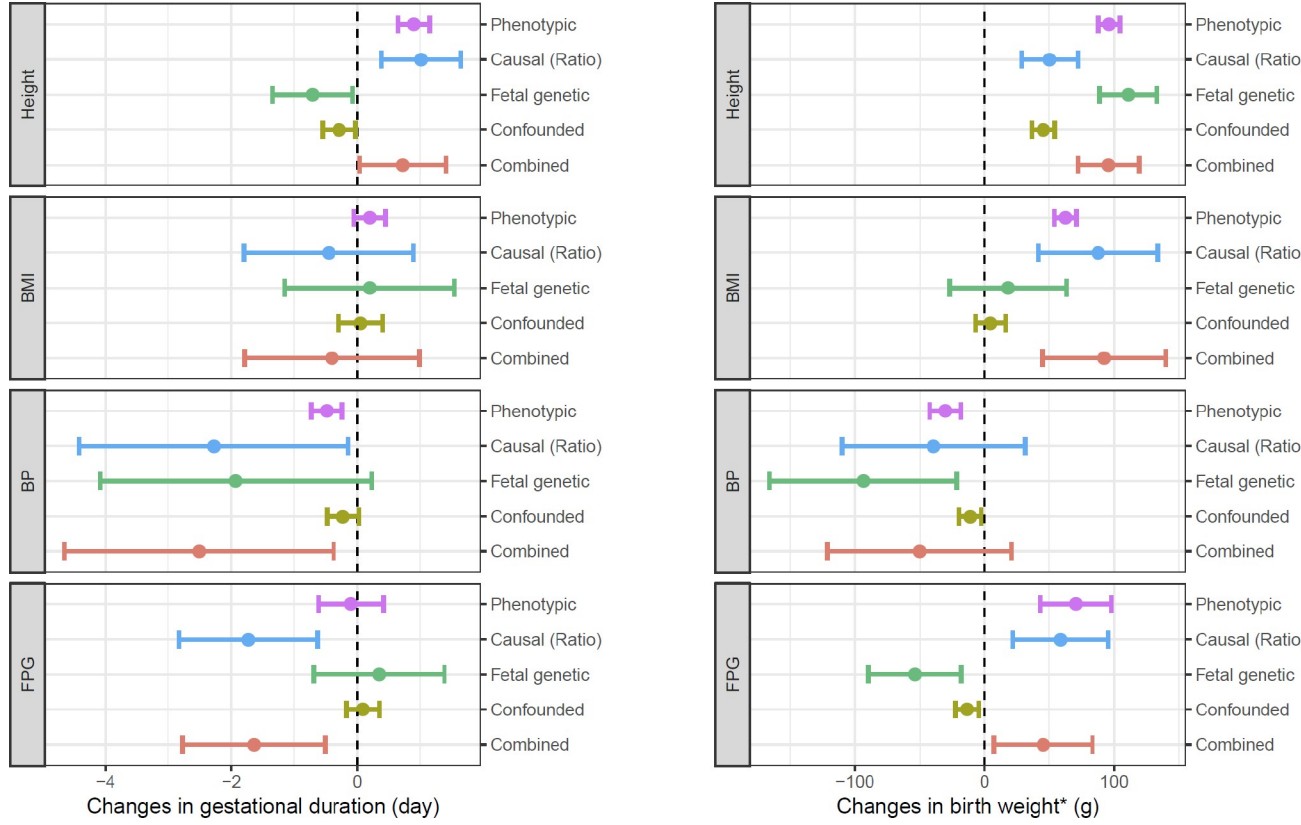

**Fig 3. Observed phenotypic associations, estimated maternal causal effects, fetal genetic effects, and genetically confounded associations per 1-SD change in maternal traits on gestational duration (left) and birth weight (adjusted by gestational duration) (right).** The 1-SD values for maternal traits are 6.4 cm (height), 4.0 kg/m$^2$ (BMI), 5.8 mmHg (BP), and 0.36 mmol/L (FPG). * indicates birth weight adjusted by gestational duration. BMI, body mass index; BP, blood pressure; FPG, fasting plasma glucose; SD, standard deviation.

between FPG and birth weight), and in these situations, the estimated maternal causal effects could be larger than the observed phenotypic associations.

## Genetically confounded associations between birth outcomes and adult phenotypes in offspring

We estimate the magnitude of genetically confounded associations between the birth outcomes and adult phenotypes (in the offspring) based on the hypothesis that the association was driven by the variants that were associated with an adult phenotype were also associated with a birth outcome (Methods). The 2 methods (Method 1 and Method 2) generated similar results (S14 Table). As shown in Fig 4 and S14 Table, a 1-SD change in birth weight (gestational-age–adjusted) was estimated to be associated with 0.20 SD (95% CI: 0.17 to 0.24, $p = 3.4 \times 10^{-28}$) and 0.076 SD (95% CI: 0.014 to 0.138, $p = 1.6 \times 10^{-2}$) differences, respectively, in adult height and BMI. One-SD increases in both gestational duration and birth weight were estimated to be associated with a 0.05 SD decrease in adult BP and a 0.025–0.03 SD decrease in adult FPG level, and birth weight was also estimated to be negatively associated with susceptibility to T2D (OR = 0.91, 95% CI: 0.85 to 0.98, $p = 7.3 \times 10^{-3}$). The genetically confounded associations between birth outcomes and adult phenotypes (S14 Fig and S15 Fig) were mainly driven by the shared genetic effects in offspring; however, the maternal effects could substantially confound the associations and point to opposite directions to the confounded associations by fetal

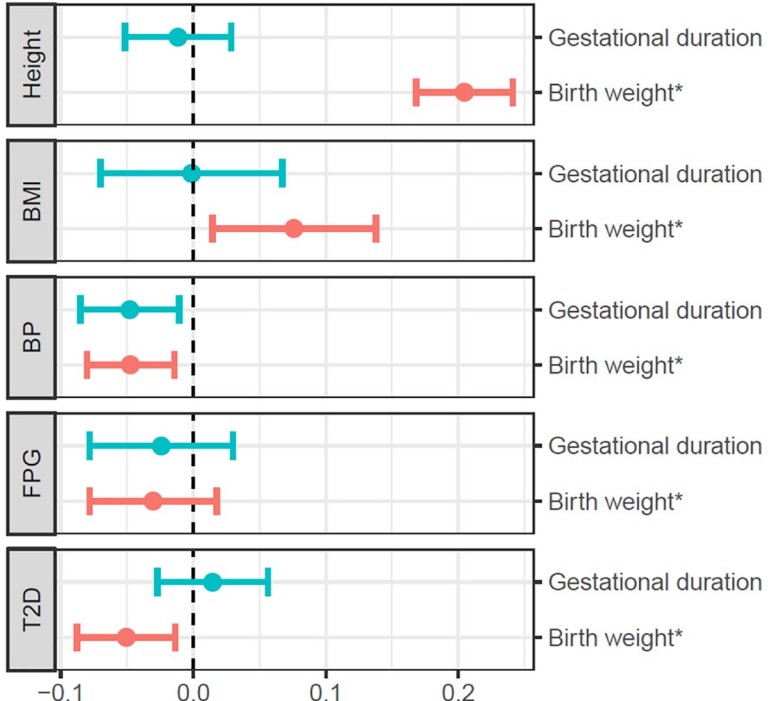

**Fig 4. Estimated differences in adult phenotypes (in SDs) per 1-SD difference in birth weight and gestational duration.** The birth weight was adjusted by gestational duration; 1 SD = 426 g. 1 SD of gestational duration was 11.4 days. The 1-SD values for adult phenotypes were assumed to be 6.4 cm (height), 4.0 kg/m$^2$ (BMI), 6.9 mmHg (BP), 0.37 mmol/L (FPG), and 1.81 for log OR of T2D. * indicates birth weight adjusted by gestational duration. BMI, body mass index; BP, blood pressure; FPG, fasting plasma glucose; SD, standard deviation; T2D, type 2 diabetes.

genetic effects in certain cases (for example, the confounded associations between gestational duration and height and between birth weight and FPG or T2D risk). We compared the genetically confounded associations between birth weight and adult phenotypes with the estimated phenotypic associations from genetic correlation [20], the observed associations in the ALSPAC data, and the reported associations from an epidemiological meta-analysis [53] (Fig 5). The estimated genetically confounded associations using our approach were similar to those estimated from genetic correlations [20] and were largely consistent with the observed or reported associations between birth weight and adult phenotypes with the exception of body height, for which the observed association (from the ALSPAC data) was significantly stronger than the estimated confounded association due to shared genetics.

## Causal effect of fetal growth on gestational duration, maternal BP, and FPG

To test the possible fetal drive of fetal growth on birth outcomes and maternal pregnancy phenotypes, we constructed genetic scores using 86 SNPs associated with birth weight with confirmed fetal effect [27] and tested their associations with birth outcomes as well as maternal BP and FPG during pregnancy (Tables 3 and 4). The fetal genetic score for birth weight was significantly associated with gestational-age–adjusted birth weight with an $R^2$ = 3.1%. The paternal transmitted haplotype score ($S_{h3}$) consistently has larger effect than the maternal transmitted score ($S_{h1}$) (S15 Table), probably because of a negative maternal effect of the same alleles on

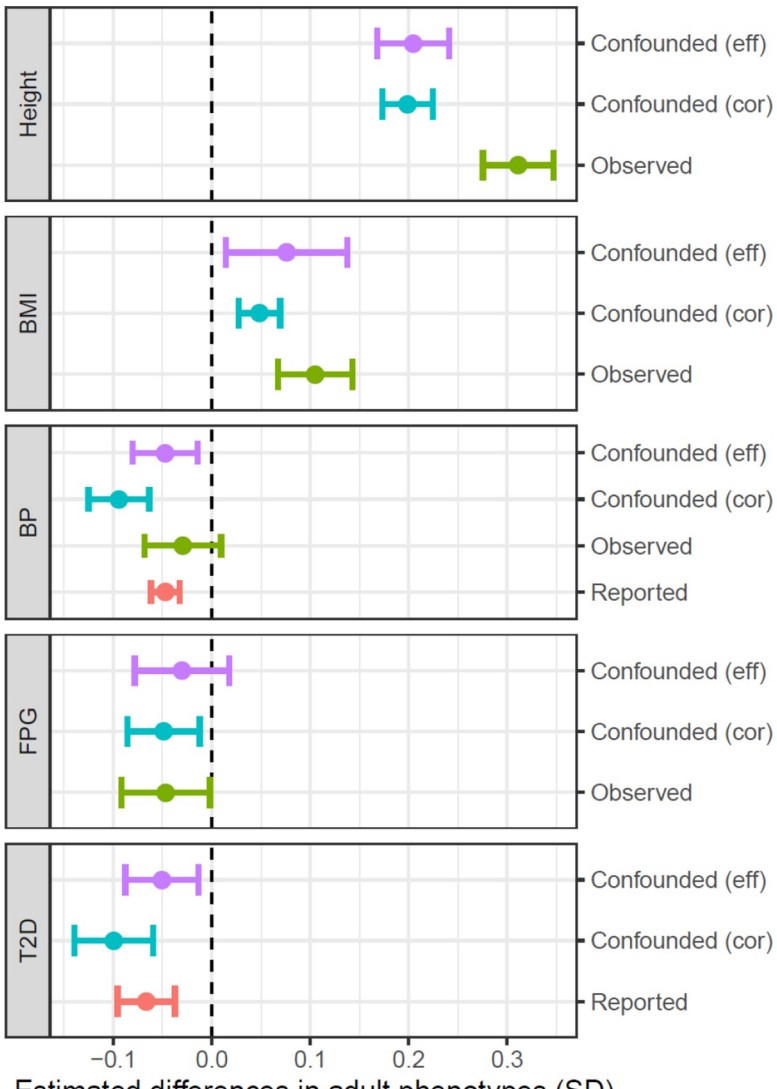

**Fig 5. Comparing the magnitudes of genetically confounded associations (differences in adult phenotypes [in SDs] per 1-SD difference in birth weight) with observed and reported phenotypic associations.** Confounded (eff): genetically confounded associations estimated based on the fetal genetic effect on birth weight (adjusted by gestational duration) of the variants associated with an adult phenotype. Confounded (cor): genetically confounded associations between birth weight (unadjusted by gestational duration) and adult phenotype based on the reported genetic correlations [20]. Observed: observed phenotypic associations between gestational-duration–adjusted birth weight and offspring height, BMI, BP, and FPG measured at age 17 in ASLPAC. Reported: the reported associations between birth weight (unadjusted by gestational duration) and BP and T2D susceptibility from a recent epidemiological study [53]. One-SD values in the gestational-duration–adjusted and unadjusted birth weight were 426 g and 484 g, respectively. BMI, body mass index; BP, blood pressure; FPG, fasting plasma glucose; SD, standard deviation; T2D, type 2 diabetes.

birth weight and birth length, as shown by the negative effect of the nontransmitted haplotype score ($S_{h2}$) (Table 3). The maternal and paternal transmitted birth weight haplotype scores ($S_{h1}$ and $S_{h3}$) were also associated with shorter gestational duration and increased preterm birth risk (Table 3 and S16 Fig). Using the paternal transmitted haplotype score as an instrument, the estimated causal effects were 3.2 days (95% CI: 1.5 to 5.0, $p = 2.7 \times 10^{-4}$) shorter gestation and an approximate doubling of the preterm birth risk per 1-SD changes in fetal growth rate (Table 3). In addition, we also observed significant associations between paternal transmitted

**Table 3. Associations between birth weight genetic scores and birth outcomes and estimated causal effects per 1-SD change in gestational-age–adjusted birth weight.**

| Genetic Score Association and Causal Estimation[a] | Gestational Days | | | Preterm Birth (log[OR]) | | | Birth Weight (g) | | | Birth Length (cm) | | |
|---|---|---|---|---|---|---|---|---|---|---|---|---|
| | Beta | SE | p-Value | Beta | SE | p-Value | Beta | SE | p-Value | Beta | SE | p-Value |
| Maternal transmitted ($\beta_{h1}$) | −0.0056 | 0.0027 | 0.035* | 0.0024 | 0.00064 | 0.00018* | 0.79 | 0.087 | $1.80 \times 10^{-19*}$ | 0.0018 | 0.00048 | 0.00022* |
| Maternal nontransmitted ($\beta_{h2}$) | 0.001 | 0.0027 | 0.7 | −0.00058 | 0.00065 | 0.37 | −0.19 | 0.088 | 0.029* | −0.00022 | 0.00049 | 0.66 |
| Paternal transmitted ($\beta_{h3}$) | −0.0099 | 0.0026 | 0.00018* | 0.0019 | 0.00064 | 0.003* | 1.3 | 0.087 | $1.70 \times 10^{-48*}$ | 0.0031 | 0.00048 | $2.30 \times 10^{-10*}$ |
| Maternal effect ($\beta_{MY}$) | 0.0026 | 0.0023 | 0.26 | $-5.80 \times 10^{-5}$ | 0.00055 | 0.92 | −0.33 | 0.076 | $1.10 \times 10^{-5*}$ | −0.00073 | 0.00042 | 0.082 |
| Fetal effect ($\beta_{FY}$) | −0.0083 | 0.0023 | 0.00029* | 0.0025 | 0.00056 | $9.20 \times 10^{-6*}$ | 1.1 | 0.075 | $1.00 \times 10^{-50*}$ | 0.0025 | 0.00042 | $1.30 \times 10^{-9*}$ |
| Causal (TSLS) | −2.92 | 0.805 | 0.00029* | 0.837 | 0.267 | 0.0017* | NA | | | 1.05 | 0.124 | $2.40 \times 10^{-17*}$ |
| Causal (ratio) | −3.24 | 0.889 | 0.00027* | 0.624 | 0.215 | 0.0036* | | | | 1 | 0.172 | $5.10 \times 10^{-9*}$ |

[a]The effect size (beta) and SEs of genetic score association were given by per unit (g) change in genetic scores; the causal effect sizes were based on per 1-SD (1 SD = 426 g) change in gestational-age–adjusted birth weight. **Abbreviations:** NA, not applicable; SD, standard deviation; SE, standard error; TSLS, two-stage least-squares.
*p-Values less than 0.05.

birth weight score ($S_{h3}$) and maternal SBP ($p = 2.2 \times 10^{-2}$) measured during pregnancy and the estimated causal effects (ratio method) were 1.4 mmHg (95% CI: 0.2 to 2.6, $p = 2.3 \times 10^{-2}$) increase in SBP per 1-SD changes in fetal growth rate (Table 4 and S17 Fig).

## Discussion

In this report, we utilized a haplotype-based genetic score approach to explicitly model the maternal and fetal genetic effects of maternal phenotypes and pregnancy outcomes (Fig 2). From the estimated maternal and fetal genetic associations (Table 2), we estimated maternal causal effects and the genetically confounded associations between maternal phenotypes and birth outcomes (Fig 3), as well as the genetically confounded associations between birth

**Table 4. Associations between birth weight genetic scores and maternal BP, glucose levels, and estimated causal effects per 1-SD changes in gestational-age–adjusted birth weight.**

| Genetic Score Association and Causal Estimation[a] | SBP[b] (mmHg) | | | DBP[b] (mmHg) | | | FPG[c] (mmol/L) | | |
|---|---|---|---|---|---|---|---|---|---|
| | Beta | SE | p-Value | Beta | SE | p-Value | Beta | SE | p-Value |
| Maternal transmitted ($\beta_{h1}$) | −0.0026 | 0.0019 | 0.17 | −0.00021 | 0.0013 | 0.87 | −0.00011 | 0.00022 | 0.61 |
| Maternal nontransmitted ($\beta_{h2}$) | 0.00057 | 0.0019 | 0.77 | 0.0006 | 0.0013 | 0.65 | −0.00025 | 0.00022 | 0.26 |
| Paternal transmitted ($\beta_{h3}$) | 0.0043 | 0.0019 | 0.022* | 0.0021 | 0.0013 | 0.1 | $8.90 \times 10^{-5}$ | 0.00022 | 0.69 |
| Maternal effect ($\beta_{MY}$) | −0.0032 | 0.0016 | 0.053 | −0.00083 | 0.0011 | 0.47 | −0.00022 | 0.0002 | 0.26 |
| Fetal effect ($\beta_{FY}$) | 0.00054 | 0.0016 | 0.74 | 0.00065 | 0.0011 | 0.57 | 0.00011 | 0.00019 | 0.55 |
| Causal (TSLS) | 1.2 | 0.551 | 0.03* | 0.641 | 0.378 | 0.09 | 0.0203 | 0.0565 | 0.72 |
| Causal (ratio) | 1.41 | 0.621 | 0.023* | 0.69 | 0.426 | 0.1 | 0.029 | 0.0724 | 0.69 |

[a]The effect size (beta) and SEs of genetic score association were given by per unit (g) change in genetic scores; the causal effect sizes were based on per 1-SD (1 SD = 426 g) change in gestational-age–adjusted birth weight.
**Abbreviations:** ALSPAC, The Avon Longitudinal Study of Parents and Children; BP, blood pressure; DBP, diastolic blood pressure; FPG, fasting plasma glucose; HAPO, Hyperglycemia and Adverse Pregnancy Outcome; SBP, systolic blood pressure; SD, standard deviation; SE, standard error; TSLS, two-stage least-squares.
[b]Maternal BP (SBP and DBP) in ALSPAC and HAPO.
[c]FPG during pregnancy measured in HAPO only.
*p-Values less than 0.05.

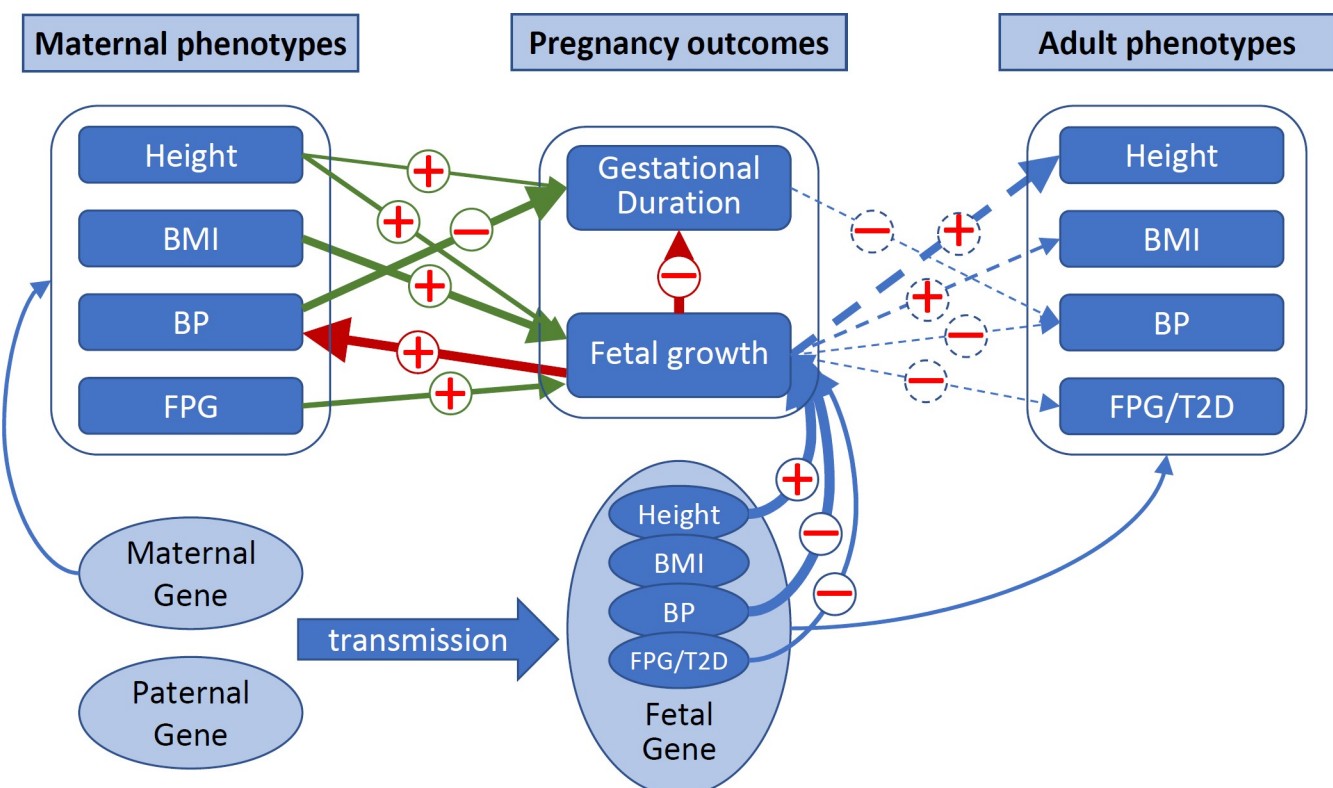

**Fig 6. Estimated maternal and fetal genetic effects underlying the associations between maternal phenotypes and pregnancy outcomes and their associations with adult phenotypes.** Blue arrows: maternal or fetal genetic effects. Green arrows: maternal causal effects. Red arrows: "fetal drive." Dashed arrows: genetically confounded associations between birth outcomes and late adult phenotypes due to shared genetics. The widths of the arrows were approximately proportional to the estimated effect sizes evaluated by per-SD changes, which can be found in S13 Table (maternal causal effect), Tables 3 and 4 (fetal drive), and Fig 4 (genetically confounded associations). BMI, body mass index; BP, blood pressure; FPG, fasting plasma glucose; SD, standard deviation; T2D, type 2 diabetes

outcomes and adult phenotypes (Fig 4). We also tested whether fetal growth (as indicated by gestational-age–adjusted birth weight) could influence gestational duration, maternal BP, and maternal FPG levels during pregnancy (Tables 3 and 4). Our results revealed complex maternal and fetal genetic effects in shaping the associations between maternal phenotypes and birth outcomes and their associations with adult phenotypes (Fig 6).

Our results support that maternal height influences the duration of gestation and fetal growth, and alleles associated with adult height also influence birth size through fetal genetic effects. These new results support our previous finding [15] with additional evidence. We observed evidence supporting causal effect of maternal BMI on birth weight and length. Similar observations have been reported before [24]. We found no evidence demonstrating that SNPs associated with BMI have significant fetal genetic effects on birth weight or length. We also observed that alleles elevating BP are associated with shorter gestational duration though a maternal effect but are associated with reduced fetal growth through a fetal genetic effect. The later finding is consistent with Horikoshi and colleagues [20]; however, it is inconsistent with Warrington and colleagues [27], who did not find a fetal effect of alleles for BP on fetal growth. We showed a positive maternal effect of FPG on fetal growth. This result is consistent with previous studies [5, 24]. However, the glucose-increasing alleles in the fetus are associated with reduced birth weight, which is in line with the epidemiological finding that paternal diabetes is associated with lower birth weight [54] and with the fetal insulin hypothesis [19]. The T2D-

associated SNPs can influence birth weight in a similar way as FPG SNPs but with a weaker maternal effect and a stronger fetal effect. These observations indicate that the maternal causal effect is mainly driven by maternal FPG levels. The fetal genetic effect is likely mediated by fetal insulin, as proposed by the fetal insulin hypothesis [19]. By utilizing the birth weight genetic score built on paternal transmitted alleles, our findings also support causal effects of fetal growth on gestational duration and maternal BP.

These findings have several implications. First, maternal size and fetal growth are important factors in defining the duration of gestation. This is demonstrated not only by the evidence supporting a causal effect of maternal height (size of the mother) on gestational duration, but also by the observation that the maternal or fetal genetic effects on fetal growth are usually associated with opposing effects on gestational duration. For example, rapid fetal growth due to either high maternal FPG or direct fetal genetic effects of growth-promoting alleles shortens gestational duration. Whether this "trade-off" between fetal growth and gestational duration is due to physical [55] or metabolic [56] constraints will require further investigation. Alleles associated with BP have negative impacts on both birth weight and gestational duration but mainly through either fetal genetics or a maternal effect, respectively.

Secondly, fetal growth (as evaluated by gestational-duration–adjusted birth weight and length) is influenced by both maternal and fetal effects. In addition to the positive maternal effect of maternal FPG on birth weight, the many alleles associated with body height, BP, FPG level, or T2D susceptibility can influence fetal growth through fetal genetic effects. The alleles associated with body height are positively associated with birth size, whereas the alleles associated with higher metabolic risks (for example, high BP, blood glucose level, and higher risk of T2D) reduce birth weight, which also suggests that lower birth weight (or, more precisely, small for gestational age status) might be a predictor of the load of genetic metabolic risks. Our results show that the shared genetic effects largely explain the life-course associations between birth weight and many cardiometabolic phenotypes (Fig 5), a result that is consistent with the reported inverse genetic correlations between birth weight and late-onset metabolic disorders [20], and both support a strong genetic rather than an environmental effect underlying the life-course association between birth weight and later metabolic risks. Compared with the previous analyses based on genetic correlations using genome-wide SNPs, our approach estimated the life-course associations by extrapolating the effects of the top GWA SNPs associated with the adult phenotypes and therefore has a specific mechanistic implication—the life-course associations were mainly driven by genetic variants with large effects on adult phenotypes rather than by the variants with large effects on fetal growth. In addition, our method can partition the genetically confounded associations to either maternal effects or shared genetic effects in offspring (S1 Text).

The results from this study also reveal a major theme in human pregnancy—both maternal effects and direct fetal genetic effects jointly shape the observed associations between maternal phenotypes and pregnancy outcomes. The same genetic variants can influence different birth outcomes or the same birth outcomes through both maternal and fetal effects, and these 2 types of effects can be antagonistic, as exemplified by the opposing maternal and fetal effects of the FPG associated alleles on birth weight, or the effects of height-associated alleles on gestational duration. These complex mechanisms can be further complicated by "fetal drive," as shown by the associations between paternal transmitted birth weight genetic score and gestational duration as well as maternal BP, i.e., fast fetal growth shortens gestational duration and increases maternal BP (Tables 3 and 4).

Our study had a number of limitations. First, there were some incomplete and heterogeneous phenotype data. Most maternal phenotypes (for example, age, height, and BMI) and birth outcomes (for example, gestational duration and birth weight) were available from the

study data sets; however, FPG level during pregnancy was only available in HAPO, and BP data were only available from HAPO and ALSPAC. Another limitation is that because of incomplete data, we were not able to include important environmental or socioeconomic factors in the analysis. Some of the maternal exposures (for example, maternal smoking) are known to be associated with both maternal phenotypes and birth outcomes [57], which may introduce confounding. However, we argue that the genetic scores used in our study are not known to be associated with these factors, and therefore, our analyses are robust to potential confounding due to these environmental factors [58]. Biological pleiotropy is always an issue in causal inference using genetic instruments, especially when a large number of genetic variants are used [45, 59]. We used the MR-PRESSO method to detect and remove SNPs with horizontal pleiotropic effects (S1 Text). The results (S16 Table and S17 Table) were essentially the same as the corresponding haplotype genetic score associations (Tables 2, 3 and 4). There is some evidence of pleiotropy (MR-PRESSO global test $p < 0.05$, S18 Table and S19 Table), suggesting the identified maternal effects may be not exclusively mediated by the targeted maternal phenotypes and the fetal effects of the SNPs on a birth outcome are not always proportional to their report effects on the adult phenotype (S4 Fig). After excluding the outliers, the corrected estimates and the $p$-values did not change substantially, and all the MR-PRESSO distortion test $p$-values were nonsignificant (S18 Table, S19 Table, and S18 Fig).

Multiple maternal and fetal phenotypes were examined in this study. The heritability (S2 Table) and the variance explained by their corresponding genetic scores (S4 Table) varied across these phenotypic traits. Because of these differences in variance being explained depending upon phenotype, we had different powers in testing the causal effects or genetic correlations among these traits. In the analysis of causal effects of fetal growth on maternal BP and FPG during pregnancy (Table 4), the phenotype data were only available in a subset of samples, which compromises the power of statistical analyses. These results will benefit from further replication in independent cohorts. Genetic variants associated with birth weight were used as genetic instrument for fetal growth. However, birth weight is an endpoint of fetal growth, and it cannot capture the temporal changes and effects of fetal growth during the course of pregnancy. Nevertheless, because the major aim of the study is to distinguish and to compare the relative contributions of maternal and fetal effects, we believe we have provided robust evidence in support of our major conclusions (Fig 6).

To conclude, our study revealed that many SNPs associated with maternal height, BP, and blood glucose levels (or T2D susceptibility) can have various maternal and fetal genetic effects on gestational duration and fetal growth. These maternal and fetal genetic effects may explain the observed associations between the studied maternal phenotypes and birth outcomes, as well as the life-course associations between these birth outcomes and adult phenotypes. Our findings related to gestational-age–adjusted birth weight suggest that rapid fetal growth might reduce gestational duration and increase maternal BP. These findings provide additional insights into the mechanisms behind the observed associations between maternal phenotype and birth outcomes and their life-course impacts on later-life health. Although our current study focused on pregnancy outcomes measured at birth (for example, gestational duration and birth weight), similar analysis can be conducted on longitudinal measures (for example, those related to growth pattern in early life and later adulthood). With the accumulation of more longitudinal birth cohorts, the dissection of maternal and fetal genetic effects may open up future opportunities to explore how maternal effect, fetal development, and genetics influence long-term health and well-being.

## Supporting information

**S1 STROBE Checklist. STROBE checklist.** STROBE, Strengthening the Reporting of Observational Studies in Epidemiology.
(PDF)

**S1 Text. Supplementary methods.**
(PDF)

**S1 Data. GWA SNPs used in constructing genetic scores.** GWA, genome-wide association; SNP, single-nucleotide polymorphism.
(XLSX)

**S1 Table. Data sets and number of samples.**
(PDF)

**S2 Table. GWA SNPs used to build the genetic scores.** GWA, genome-wide association; SNP, single-nucleotide polymorphism.
(PDF)

**S3 Table. Association between maternal phenotypes and birth outcomes.**
(PDF)

**S4 Table. Associations between maternal genetic scores and maternal traits.**
(PDF)

**S5 Table. Associations between maternal height genetic scores and maternal height.**
(PDF)

**S6 Table. Associations between maternal BMI genetic scores and maternal BMI.** BMI, body mass index.
(PDF)

**S7 Table. Associations between maternal BP genetic scores and maternal BP (SBP and DBP).** BP, blood pressure; DBP, diastolic BP; SBP, systolic BP.
(PDF)

**S8 Table. Associations between maternal FPG and T2D genetic scores and maternal FPG.** FPG, fasting plasma glucose; T2D, type 2 diabetes.
(PDF)

**S9 Table. Correlation (p-value) between maternal genotype score ($S_{mat}$), fetal genotype score ($S_{fet}$), and the 3 haplotype genetic scores: $S_{h1}$, $S_{h2}$, and $S_{h3}$.**
(PDF)

**S10 Table. Association between haplotype genetic scores and birth outcomes—meta-analyses using random-effects model.**
(PDF)

**S11 Table. Association between haplotype genetic scores and birth outcomes based on the ALSPAC data set only.** ALSPAC, The Avon Longitudinal Study of Parents and Children.
(PDF)

**S12 Table. Association between haplotype genetic scores and birth outcomes based on meta-analysis of the FIN, MoBa, DNBC, HAPO, and GPN data sets.** DNBC, The Danish National Birth Cohort; FIN, The Finnish birth data set; GPN, The Genomic and Proteomic Network for Preterm Birth Research; HAPO, Hyperglycemia and Adverse Pregnancy

Outcome; MoBa, The Mother Child data set of Norway.
(PDF)

**S13 Table. Phenotypic associations, estimated maternal causal effects, and genetically confounded associations (per 1-SD changes in maternal traits) between maternal traits and birth outcomes.** SD, standard deviation.
(PDF)

**S14 Table. Estimated changes in adult phenotypes per 1-SD changes in birth outcomes.** SD, standard deviation.
(PDF)

**S15 Table. Association between fetal birth weight genetic score and gestational-age–adjusted birth weight (gram).**
(PDF)

**S16 Table. MR-PRESSO: Effects of maternal traits on birth outcomes.** MR-PRESSO, mendelian randomization pleiotropy residual sum and outlier.
(PDF)

**S17 Table. MR-PRESSO: effects of fetal growth on birth outcomes (A) and maternal BP and glucose levels (B).** BP, blood pressure; MR-PRESSO, mendelian randomization pleiotropy residual sum and outlier.
(PDF)

**S18 Table. MR-PRESSO global test, outlier test, and distortion test results: effects of maternal traits on birth outcomes.** MR-PRESSO, mendelian randomization pleiotropy residual sum and outlier.
(PDF)

**S19 Table. MR-PRESSO global test, outlier test, and distortion test results: effects of fetal growth on birth outcomes (A) and maternal BP and glucose levels (B).** BP, blood pressure; MR-PRESSO, mendelian randomization pleiotropy residual sum and outlier.
(PDF)

**S1 Fig. Processing of phenotype and genotype data.**
(PDF)

**S2 Fig. General representation of confounded association.**
(PDF)

**S3 Fig. Genetically confounded association between a maternal phenotype and a birth outcome (A) and between a birth outcome and an adult phenotype in offspring (B) because of shared genetics**.
(PDF)

**S4 Fig. Multivariable MR analysis of maternal and fetal genetic effect.** MR, mendelian randomization.
(PDF)

**S5 Fig. Distributions of maternal height, weight, and BMI.** BMI, body mass index.
(PDF)

**S6 Fig. Distributions of maternal BP and glucose levels.** BP, blood pressure.
(PDF)

**S7 Fig. Distributions of gestational days, birth weight, and birth length.**
(PDF)

**S8 Fig. Estimated effect sizes of adult height haplotype scores on pregnancy outcomes.**
(PDF)

**S9 Fig. Estimated effect sizes of BMI haplotype scores on pregnancy outcomes.** BMI, body mass index.
(PDF)

**S10 Fig. Estimated effect sizes of BP haplotype scores on pregnancy outcomes.** BP, blood pressure.
(PDF)

**S11 Fig. Estimated effect sizes of blood glucose haplotype scores on pregnancy outcomes.**
(PDF)

**S12 Fig. Estimated effect sizes of T2D haplotype scores on pregnancy outcomes.** T2D, type 2 diabetes.
(PDF)

**S13 Fig. Estimated sizes of causal effects and genetically confounded associations per 1-SD changes in maternal traits on preterm birth risk and birth length.** SD, standard deviation.
(PDF)

**S14 Fig.  Estimated differences in adult phenotypes (SD) per 1-SD difference in gestational days (left) and preterm birth risk (right). SD, standard deviation**.
(PDF)

**S15 Fig.  Estimated differences in adult phenotypes (SD) per 1-SD difference in gestational-age–adjusted birth weight (left) and birth length (right). SD, standard deviation**.
(PDF)

**S16 Fig. Estimated effect sizes of birth weight haplotype scores on pregnancy outcomes.**
(PDF)

**S17 Fig. Estimated effect sizes of birth weight haplotype scores on maternal BP.** BP, blood pressure.
(PDF)

**S18 Fig. An example of multivariable MR estimate.** MR, mendelian randomization.
(PDF)

## Acknowledgments

We thank the participants in the Finnish birth cohort as well as the research group who collected the data. We are grateful to all the participating families in Norway who took part in the MoBa cohort study. We are extremely grateful to all the families who took part in ALSPAC, the midwives for their help in recruiting them, and the whole ALSPAC team, which includes interviewers, computer and laboratory technicians, clerical workers, research scientists, volunteers, managers, receptionists, and nurses. We are very grateful to all DNBC families who took part in the study. We would also like to thank everyone involved in data collection and biological material handling. We would like to acknowledge the participants and research personnel at the participating HAPO field centers. We thank the infants and their parents who agreed to take part in the GPN study and the medical and nursing colleagues who collected that data.

We thank dbGAP for depositing and hosting the phenotype and genotype data of the DNBC, HAPO, and GPN data sets.

## Author Contributions

**Conceptualization:** Louis J. Muglia, Ge Zhang.

**Data curation:** Jonas Bacelis, Amit Srivastava, Amy Rouse, Mikko Hallman, Ge Zhang.

**Formal analysis:** Jing Chen, Ge Zhang.

**Funding acquisition:** Louis J. Muglia, Ge Zhang.

**Investigation:** Jing Chen, Rachel M. Freathy, Deborah A. Lawlor, Jeffrey C. Murray, Scott M. Williams, Bo Jacobsson, Louis J. Muglia, Ge Zhang.

**Methodology:** Jing Chen, Jonas Bacelis, Bjarke Feenstra, Ge Zhang.

**Project administration:** Bo Jacobsson, Louis J. Muglia, Ge Zhang.

**Resources:** Jonas Bacelis, Mikko Hallman, Kari Teramo, Mads Melbye, Bjarke Feenstra, George Davey Smith, Deborah A. Lawlor, Jeffrey C. Murray, Bo Jacobsson, Louis J. Muglia, Ge Zhang.

**Software:** Jing Chen, Ge Zhang.

**Supervision:** Bo Jacobsson, Louis J. Muglia, Ge Zhang.

**Validation:** Jonas Bacelis, Pol Sole-Navais, Julius Juodakis, Ge Zhang.

**Visualization:** Jing Chen, Rachel M. Freathy, Jeffrey C. Murray, Ge Zhang.

**Writing – original draft:** Jing Chen, Rachel M. Freathy, Louis J. Muglia, Ge Zhang.

**Writing – review & editing:** Jing Chen, Jonas Bacelis, Pol Sole-Navais, Amit Srivastava, Julius Juodakis, Mikko Hallman, Kari Teramo, Bjarke Feenstra, Rachel M. Freathy, George Davey Smith, Deborah A. Lawlor, Jeffrey C. Murray, Scott M. Williams, Bo Jacobsson, Louis J. Muglia, Ge Zhang.

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
