## [Decision Letter · Decision Letter 0]

24 Feb 2020

Dear Dr. Zhang,

Thank you very much for submitting your manuscript "Haplotype genetic score analysis in 10,734 mother/infant pairs reveals complex maternal and fetal genetic effects underlying the associations between maternal phenotypes, birth outcomes and adult phenotypes" (PMEDICINE-D-19-03326) for consideration at PLOS Medicine. 

Your paper was discussed among the editorial team and sent to independent reviewers, including a statistical reviewer. The reviews are appended at the bottom of this email and any accompanying reviewer attachments can be seen via the link below:

[LINK]

In light of these reviews, we will not be able to accept the manuscript for publication in the journal in its current form, but we would like to invite you to submit a revised version that fully addresses the reviewers' and editors' comments. You will recognize that we cannot make a decision about publication until we have seen the revised manuscript and your response, and we expect to seek re-review by one or more of the reviewers. 

We hope to receive your revised manuscript by Mar 16 2020 11:59PM. Please email us (plosmedicine@plos.org) if you have any questions or concerns.

Please let me know if you have any questions. Otherwise, we look forward to receiving your revised manuscript in due course. 

Sincerely,

Richard Turner PhD, for Caitlin Moyer, Ph.D.

rturner@plos.org

To the data statement, we suggest adding a note to the effect that all data derived from the present study are presented with or in the paper (assuming this is the case).

Please substitute a non-declarative title, and add a study descriptor (e.g., "...: a mendelian randomization study"). 

Please convert your abstract into a three-section structure (background/methods and findings/conclusions). 

Please include summary demographic information for study participants in your abstract. 

We ask you to quote illustrative quantitative elements of your findings in the abstract to support the conclusions drawn. 

The final sentence of the "methods and findings" subsection of the abstract should summarize the study's main limitations. 

Please present conclusions in the general form "In this study, we found that ... " or similar. 

After the abstract, we will need to ask you to add a new and accessible "author summary" section in non-identical prose. You may find it helpful to consult one or two recent research papers published in PLOS Medicine to get a sense of the preferred style. 

Early in the methods section of your main text, please state whether the study had a protocol or prespecified analysis plan, and if so add the relevant document(s) as a supplementary file, referred to in the text. Please highlight analyses that were not prespecified. 

Please add a sentence to your methods section to the effect that specific ethics approval was not required for the present study (assuming this is the case). 

Where available, please quote p values alongside 95% CI throughout the paper. 

Please quote exact p values or p<0.001, unless there are specific reasons to quote smaller p values.

Throughout the text, please adapt reference call-outs to the following format: "... cardiovascular diseases [11,12].".

In your reference list, please ensure that journal names are abbreviated consistently, e.g., "J Pediatr." for reference 4 and "JAMA" for reference 9. 

Please spell out the group name in reference 5. 

Please add full access details for references 31 & 40. 

Please add a completed checklist for the most appropriate reporting guideline, which we suspect will be STROBE, as a supplementary file (referred to in your methods section). In the checklist, individual items should be referred to by section (e.g., "Methods") and paragraph number rather than by line or page numbers, as the latter generally change in the event of publication. 

Comments from the reviewers:

*** Reviewer #1: 

The authors present a very interesting study on quantifying the causal effects of maternal and fetal traits on birth outcomes using genetic scores constructed haplotype genetic scores which combines the maternal and fetal effects as a single unit. Using a large number of mother/infant duos singleton of European ancestry - they were able to identify the causal effects of maternal height, pre-pregnancy BMI, blood pressure and blood glucose on pregnancy outcomes (gestational duration), birth weight and birth length. What the authors have shown is mechanistic insight into the observed association between maternal phenotypes and birth outcomes which has implications for life-course association birth outcomes and future adult outcomes. 

The results in itself are not particularly surprising that there are observed causal influences of maternal phenotypes on birth outcomes as this has been shown in many previous studies, including the authors own previous work but what is the novelty on this paper is actually the methodology itself. The authors developed a method which explicitly models maternal and fetal genetic effects using as a single analytic unit using instrumental variables of haplotype genetic scores. This means that a key advantage of this method is that they could explore whether any effects on fetal outcomes are associated gestational duration and whether fetal growth has an effect on gestational duration as well as maternal phenotypes in pregnancy. This is a complex analysis but presented in a way that is intuitive and generally easy to follow. The statistical methods underpinning all the analysis are a series of linear regression models, which have additive in nature, and then overall results are pooled across cohorts using meta-analyses

There are few comments I have which the authors could find useful:

Abstract - Suggest conforming to PLOS Medicine style of three sections - background/methods&findings/conclusion. The methods are substantial so it would be useful to in the abstract to summarise the methodology and study design

The study is rationale is strong and extends previous work by others and the authors themselves in determining causal associations between maternal phenotypes and pregnancy outcomes and adult cardio metabolic diseases. There is a good body of evidence already that the effects of maternal characteristics on fetal outcomes. The authors extend this body of evidence by combing maternal/fetal as a single unit to model genetic scores using a haplotype genetic score as an instrumental variable. Figure 2 is useful, the models assumes paternal non-transmitted alleles h4 (which isn't described in the legend) has no bearing on the maternal or fetal affects. This seems like a logical assumption but is there any evidence to suggest that paternal non-transmitted phenotypes may influence maternal phenotypes? 

Methods - ALSPAC blood pressure measurements: authors used BP measurements between 30-36 weeks - was this an average or a single point estimate?

Methods - PCA analysis to exclude non-European ancestry - how many were excluded in the end from non-European ancestry and how did this map up to what was self-reported as ethnicity? 

Methods - genotype data - From previous GWA (Table S4 provided) - it would appear that heritability of certain most of the traits investigated varies from high levels (height - H2 0.8) to low (birthweight H2 0.3). This makes logical sense as features such as birth weight are have strong environmental influences (i.e. smoking in pregnancy). Can the authors discuss how the variation in the GRS instruments may have on the overall results and how better instruments would explain higher levels of variation in the phenotypic traits. 

Methods - association tests between haplotype genetic scores and pregnancy outcomes. How were potential interactions between handled or identified between gene-gene or gene-environment?

Methods - multivariate MR analysis - when combing the results from all datasets, given the heterogeneity in the datasets (observed in the distributions provided in supplemental) as well as significant heterogeneity identified in several of the meta-analyses by Cochran's Q (Supplemental Figures S8-S12), it would have made sense to use a random-effects model. At least this should be considered in meta-analyses where there is a large amount of heterogeneity. This does have implications on the overall findings, but it would test the robustness of the pooling process. 

Results - association between genetic score and maternal phenotypes. The genetic scores explain > 20% of the maternal height variance but the remaining genotypes explain less than 5% variance of the maternal phenotypes - suggesting strong environmental influences which is not surprising. This also suggests that perhaps in the haplotype model exploring casual effects using the genetic instruments to consider incorporating more environmental covariates (presumably some of the birth cohorts would have collected these data - for instance ALSPAC). Though genetic risk scores may not associated with many of the environmental demographic factors, they might have some association with potential other maternal phenotypes indicative of maternal cardio-metabolic health which may have not been considered

Results - one of the limiting factors which have not been considered is maternal nutrient intake during pregnancy, and certain behavioural factors (i.e. smoking, alcohol usage) which may have some link to genetic link towards to these behaviours. Could the authors comment whether these behavioural or nutrition factors could have a large influence on their findings. 

*** Reviewer #2: 

The proposed manuscript entitled "Haplotype genetic score analysis in 10,734 mother/infant pairs reveals complex maternal and fetal genetic effects underlying the associations between maternal phenotypes, birth outcomes and adult phenotypes" by Chen et al. seeks to dissect the maternal and fetal genetic effects and their complex relationship between maternal traits and offspring outcomes including gestational duration, birth weight, and birth length. They utilized haplotype scores and Mendelian randomization to identify that tall maternal stature and high maternal blood glucose casually increase birth size, in the neonate, height and blood pressure increasing alleles can lead to increased and decreased birth weight, alleles associated with increased birth weight reduce the time of gestation and increase maternal blood pressure. This is an interesting article with some intriguing results but there are several issues that first need to be addressed.

1) The Abstract needs to be expanded on and fully elucidated for the journal and should include Background, Methods, and Findings and Conclusions. Also given the complexity of some of the underlying statistical genetics in the article it should include an Author Summary as well to better present this to a clinical audience. 

2) The second paragraph in the introduction describes a lot of the recent work on the underlying maternal/fetal genetic relationship with birthweight but provides less information on the relationship between birthweight and some of the other fetal outcomes proposed here. This paragraph should be shortened and go into more detail regarding the latter as this is the major gap in the literature this paper is seeking to address.

3) Throughout the manuscript when you discuss blood pressure it is unclear how this is defined. Are you using systolic or diastolic blood pressure as both are included in the supplementary tables or are you using a combined phenotype? 

4) The authors used the program MR-PRESSO to avoid horizontal pleiotropy as the correctly mention the high correlation between traits. However, they do not provide any measure of horizontal pleiotropy. As the authors of the program mention, it is only useful when horizontal pleiotropy is < 50%. This should be addressed in some way in the manuscript. This is not clearly shown in the supplementary tables either. 

5) For the meta-analysis was any specific computer program used and are the Q scores provided in any of the supplementary tables?

6) The inclusion of the fasting blood glucose in ALSPAC 18 years later seems spurious, as most blood chemistry would be expected to change over time and not sure how this would impact any fetal measurements. Horse has already left the barn in this case. 

7) In the results, the fact that there is a smaller sample size (2 cohorts) it would appear unsurprising that the blood pressure is half, also given that there is a potential age discrepancy between the two and blood pressure is notoriously heterogeneous.

8) How is pre-term birth being assessed throughout, there is no clear definition provided but the term is used throughout the results. 

9) I would be very cautious regarding the fasting glucose results as this only comes from one cohort, so I am not sure there is strong evidence for the results here.

10) In the meta-analysis was any sensitivity testing done, such as drop one cohort out and see if they have more influence on the overall results. 

*** Reviewer #3: 

This paper by Chen, Bacelis, et al. explores the contributions of maternal and fetal genetics, and maternal phenotypes, to pregnancy outcomes - a topic that has some significant general interest. Since I am not a statistical geneticist my critique is brief and based more on common sense than on technical considerations. Nonetheless, it might be useful since PLoS Medicine's target audience is broad. 

This is an extremely complex piece of work, involving phenotypic (height, weight, metabolic, pregnancy duration, etc.) and genotypic (SNP) data of 10,734 mother/infant pairs from six independently conducted birth studies. Each of these birth studies had different designs and different ways of obtaining both the genotypic and phenotypic data - which necessitated several significant compromises in their joint analysis. To cite just one example, maternal fasting plasma glucose (FPG) levels during pregnancy were available only in the HAPO study, while FPG was measured in over 4000 ALSPAC mothers not during pregnancy but in a follow-up data collection 18 years after the pregnancy.

I believe the paper needs to be rewritten, and I have two suggestions:

1. The concepts of maternal and fetal "genotype genetic scores" and "haplotype genetic scores" will be new for a general readership and are important for understanding the study. These concepts should be illustrated diagrammatically in a main figure. The current diagrams with arrows indicating potential maternal-fetal interactions are useful and should be kept - but should simplified to deal with only the strongest and most reproducible findings (see #2 below).

2. More crucially, what are the strongest and most reproducible findings in this study? Because of the way the authors chose to analyze and describe all 6 birth studies in parallel, it is difficult or impossible to extract a clear notion of replication of any of the specific findings across these independently conducted and differently designed birth studies. The authors should consider analyzing one of the birth studies (whichever one they see as most complete for their analytical approach and for the maternal and fetal phenotypes that they believe are most informative), and then vetting their strongest findings using the other 5 studies as replication sets. This approach would be easier to grasp and would have obvious advantages, both for distilling the most convincing results and for simplifying the tables, which are currently very difficult to read. The "study pooling" approach would not have to be totally discarded - it could be presented as supplemental data.

***

[LINK]

---

## [Decision Letter · Decision Letter 1]

22 May 2020

Dear Dr. Zhang,

Thank you very much for submitting your revised manuscript "Dissecting maternal and fetal genetic effects underlying the associations between maternal phenotypes, birth outcomes and adult phenotypes – a haplotype-based genetic score analysis in mother/child pairs" (PMEDICINE-D-19-03326R1) for consideration at PLOS Medicine. 

Your paper was re-evaluated by a senior editor and discussed among all the editors here. It was also discussed with an academic editor with relevant expertise, and sent to three independent reviewers, including a statistical reviewer. The reviews are appended at the bottom of this email and any accompanying reviewer attachments can be seen via the link below:

[LINK]

In light of these reviews, I am afraid that we will not be able to accept the manuscript for publication in the journal in its current form, but we would like to consider a revised version that addresses the reviewers' and editors' remaining comments. Obviously we cannot make any decision about publication until we have seen the revised manuscript and your response, and we may seek re-review by one or more of the reviewers. 

We expect to receive your revised manuscript by May 29 2020 11:59PM. Please email us (plosmedicine@plos.org) if you have any questions or concerns.

We look forward to receiving your revised manuscript. 

Sincerely,

Caitlin Moyer, Ph.D.

Associate Editor 

PLOS Medicine

plosmedicine.org

1.Response to editor comments: “Please quote exact p values or p<0.001, unless there are specific reasons to quote smaller p values.”

Your response: We intend to keep the actual exact P values. We understand that expressing P values to more than 3 significant digits usually does not add useful information because P values with extreme results are sensitive to biases or departures from the assumptions of the test. However, in this manuscript, exact P values were used to compare the relative magnitude of the effects of the three haplotype scores tested using the same statistical model.

After consulting with the statistical reviewer, we ask that you please explain this further. Our understanding is that in most cases p-values shouldn’t be directly compared, except perhaps in the context of models where the sample sizes are exactly the same. Can you please clarify the rationale behind direct comparison of p-values and note in the relevant section of the Methods where this is applied to your analyses.

2. Response to Reviewer comments: Reviewer 1, point 9: Would you please address this point a bit further, particularly given the well accepted relationship between maternal smoking and fetal effects?

3. Please revise your title according to PLOS Medicine's style. Your title must be nondeclarative and not a question. It should begin with the main concept if possible. Please place the study design ("A randomized controlled trial," "A retrospective study," "A modelling study," etc.) in the subtitle (ie, after a colon). We suggest incorporating that the study included Mendelian randomization as well. We suggest: “Dissecting maternal and fetal genetic contributions to the associations between maternal phenotypes, birth outcomes and adult phenotypes: a Mendelian randomization and haplotype-based genetic score analysis in 10,734 mother-infant pairs of European ancestry” or similar.

In the title (and throughout), we suggest using a hyphen for “mother-infant” or “mother-child” pairs.

4. Abstract: Background: We suggest revising the last sentence to “The causal mechanisms and the relative contributions of maternal and fetal genetic effects behind these observed associations are unresolved.”

5. Abstract: Methods and Findings: We suggest revising to: Please clarify “these observed associations” as it is not clear what this is referencing.

6. Abstract, and throughout: Please revise to avoid causal language, for example in the following sentence. “The maternal non-transmitted haplotype score for height was significantly associated with gestational duration (P=0.00022) and preterm birth (P=0.00097) confirming the causal effect of maternal height on gestational duration.” Similarly, please soften ‘maternal and fetal genetic effects’ to ‘maternal and fetal genetic contributions’ or similar, to avoid causal implications.

7. Abstract, Methods and Findings: If scientific notation is used for p values, please use consistently throughout.

8. Abstract, Methods and Findings: we suggest revising “...however, the glucose-increasing alleles in the fetus reduced birth weight…” to “glucose-increasing alleles in the fetus were associated with reduced birth weight…” or similar, to avoid inferring causality.

9. Abstract, Conclusions (and throughout the manuscript): Please revise this paragraph to remove language that implies causality, such as “both maternal height and fetal growth affect the duration of gestation” . Refer to associations instead.

10. Author Summary: Please place the author summary between the abstract and the introduction sections.

11. Author Summary: What did the researchers do and find?: Please reduce causal language here.

12. Author Summary: What do these findings mean?: Please reduce causal language, particularly in the second bullet point. For the first bullet point, we suggest revising to: “In this study, we observed that maternal size and fetal growth are important factors in shaping the duration of gestation.” or similar. 

13. Methods: Construction of genetic scores paragraph on page 6: Please refer to “association” rather than “effect” to avoid causal implications: “To examine the effect of fetal growth (as proxied by birth weight) on pregnancy outcomes and maternal blood pressure and FPG…”

14. Results: Please provide p values associated with all analyses. For example, on page 15, please provide p values associated with the OR for the association between birth weight and susceptibility to T2D (OR = 0.95, 95% CI: 0.92 to 0.99).

15. Discussion: At the beginning of the discussion, please summarize your findings in paragraph form rather than as a numbered list.

16. Discussion: Please temper the causal language. Under the first paragraph, in point 1, we suggest revising the final sentence to: “These new results support our previous finding [15] with additional evidence.”

17. Discussion: Final paragraph: We suggest revising this sentence to: “These maternal and fetal genetic effects may explain the observed associations between the studied maternal phenotypes and birth outcomes as well as the life-course associations between these birth outcomes and adult phenotypes.”

18. Tables: Please define abbreviations for “OR” and “se” in the legends.

19. Figure 3, 4, 5, 6: Please define abbreviations for FPG, BMI, BP, T2D in the figure legend.

20. Supporting Information Tables: Please define all abbreviations, such as for SBP, DBP, FPG, BMI, se, sd in the legends, where appropriate

Comments from the reviewers:

Reviewer #1: The authors have done a thorough job revising their manuscript. Appreciate the thought given to the responses and providing additional analyses. Given the number of the figures and tables already included and supplemental analyses, I would try to refrain any more further analyses as there already 5-6 key findings from the analyses. As with these types of studies, there could be even more analyses and permutations but it would be advisable at this stage to keep this to a minimum. I think any further analyses could be highlighted more as future research in the discussion. The only point I would add is that, perhaps in the discussion is that although the analysis was done as a cross-sectional study, the nature of the data used involves several birth cohorts which longitudinal data which opens up some future opportunities to explore later growth outcomes, including the relationship between glucose raising alleles on lower birth weight and how this relates to rapid growth in infancy (as we do observe LBW children tend to exhibit a period of rapid catch up growth as well in their first 1-2 years of life). This interplay between genetics, fetal growth, low birth rate, and subsequent growth in the first years of life would have potential to be answered using these same datasets the authors have collated. Food for thought in future exploration in more longitudinal designs. 

Reviewer #2: I thank the authors for their careful consideration of the previous comments regarding the proposed manuscript. I feel that they addressed the majority of my concerns in the updated manuscript. I do have a few minor suggestions.

I feel that the Methods/Results section is very results heavy. There is a single sentence form methods and the rest is dedicated to results. However, I feel given the complexity of the methodology a few more details would be welcome here. I think you can focus on the main findings rather than all of the results. 

Also, there is not Author Summary included as indicated in their response. I think this would be useful for a clinical audience. 

Reviewer #4: All my comments have been adequately addressed.

[LINK]

---

## [Decision Letter · Decision Letter 2]

9 Jul 2020

Dear Dr. Zhang,

Thank you very much for re-submitting your manuscript "Dissecting maternal and fetal genetic effects underlying the associations between maternal phenotypes, birth outcomes and adult phenotypes: a Mendelian randomization and haplotype-based genetic score analysis in 10,734 mother-infant pairs" (PMEDICINE-D-19-03326R2) for review by PLOS Medicine.

I apologize with the delay in returning a decision on your manuscript. I have discussed the paper with my colleagues and the academic editor and it was also seen again by one of the reviewers. I am pleased to say that provided the remaining editorial and production issues are dealt with we are planning to accept the paper for publication in the journal.

[LINK]

If you have any questions in the meantime, please contact me (cmoyer@plos.org) or the journal staff at plosmedicine@plos.org. 

We look forward to receiving the revised manuscript by Jul 16 2020 11:59PM. 

Sincerely,

Caitlin Moyer, Ph.D.

Associate Editor 

PLOS Medicine

plosmedicine.org

Requests from Editors:

1.Abstract: Methods and findings: Please spell out the abbreviation for BMI at the first use.

2.Abstract: Methods and findings: Please revise to clarify this sentence, we suggest: “Both maternal and paternal transmitted blood pressure scores were negatively associated with birth weight with a significant fetal effect (P=9.4E-3); while blood pressure alleles were significantly associated with gestational duration and preterm birth through maternal effects (P=3.3E-2 and P=4.5E-3, respectively).”

3.Abstract: Conclusions: Please change “fetus” to “the fetus” in “...alleles raising birth weight in fetus are associated…”

4.Abstract: Conclusions: Please change “...can largely explain” to “...may explain…”

5.Author summary: “What did the authors do and find?”: Please revise the second bullet point to: 

--Genetically-elevated maternal height is associated with the longer gestational duration and larger birth size. In the fetus, alleles associated with adult height are positively associated with birth size.

6.Author Summary: “What do these findings mean?” Please change “In fetus” to “In the fetus” in the second bullet point.

7.Methods: Please reference the specific supporting information file/s where these are described: “A detailed description of these data sets can be found in the Supplementary Methods.”

8.Methods: Under “Construction of Genetic Scores”: Please revise “For FPG, we used 22 SNPs associated with FPG levels identified in non-diabetic individuals” to “FPG levels identified in individuals without diabetes”

9.Results: second paragraph: Please also present the results with CIs/ p values for maternal BMI associations with gestational duration or preterm birth risk even if they did not reach statistical significance.

10.Results: third paragraph: Please present the results (with CIs and p values) for the associations observed in HAPO, between maternal blood pressure and gestational duration and birth weight.

11.Results: bottom of page 14: It seems part of the results are presented with p values given in scientific notation, and some are not. Please be consistent with the formatting.

12.Discussion: First sentence: “...to explicitly model the maternal and fetal genetic effects.” Can you please clarify this sentence- effects of/on what?

13.Discussion: Second paragraph, first sentence: To mitigate the causal language, please revise to: “Our results support that maternal height influences the duration of gestation and fetal growth, and alleles associated with adult height also influence birth size through fetal genetic effects.” Similarly, please revise the last sentence of that paragraph: By utilizing the birth weight genetic score built on paternal transmitted allele, our findings also support causal effects of fetal growth on gestational duration and maternal blood pressure.”

14.Discussion: Third paragraph, third sentence: Please revise to: “...demonstrated not only by the evidence supporting a causal effect of maternal height (size of the mother) on gestational duration, but also…”

15.Discussion: Please revise this sentence in the concluding paragraph to better reflect your findings: “Our findings related to gestational age-adjusted birth weight suggest that rapid fetal growth might reduce gestational duration and increase maternal blood pressure.” or similar.

16.Grants support: Please remove this section from the manuscript body, and ensure that all relevant information is included in the sections Competing Interests, Financial Disclosures, and Data Availability.

17.Figure 1 legend: In point 4, “Fetal” should be “fetal”

18.Figure 3 legend: Please describe the asterisk shown with fetal birth weight.

19.Supplementary information file: For the “Genotype Data” you refer to lists of SNPs published as supporting information in other publications- if you are able to, please include supporting information tables with these data in the event that files from other publications are inaccessible.

20.Supplementary Table S1 and S2: Please incorporate these into the main paper. 

21.Supplementary Figure S1: Please define “QC” in the legend

22.Supplementary Figure S2: If possible, please elaborate on what is demonstrated by this figure.

23.Supplementary Figure S5, S6, and S7: Please increase the font size on the histogram axes, it is difficult to read them. Please explain in the legend why FPG has an asterisk in Figure S6.

24.Supplementary Figure S8-S12: Please increase font sizes slightly.

25.Supplementary Figure S16 and S17: Please increase the font sizes slightly.

26.Supporting information file: Please replace "subject" with participant, patient, individual, or person.

27.Supporting information file: Under “Blood glucose and TD susceptibility”: Please replace “96,496 non-diabetic individuals” with “96,496 individuals without diabetes”

Comments from Reviewers:

Reviewer #1: I have re-reviewed the response by the authors and manuscript detail and agree that the authors have detailed their rationale now for including exact p-values. This is appropriate for the context of their analysis and interpretation of the findings.

[LINK]

---

## [Editor Report · Decision Letter 3]

21 Jul 2020

Dear Dr. Zhang, 

On behalf of my colleagues and the academic editor, Dr. Fasil Tekola-Ayele, I am delighted to inform you that your manuscript entitled "Dissecting maternal and fetal genetic effects underlying the associations between maternal phenotypes, birth outcomes and adult phenotypes: a Mendelian randomization and haplotype-based genetic score analysis in 10,734 mother-infant pairs" (PMEDICINE-D-19-03326R3) has been accepted for publication in PLOS Medicine. 

PRODUCTION PROCESS

PRESS

PROFILE INFORMATION

Thank you again for submitting the manuscript to PLOS Medicine. We look forward to publishing it. 

Best wishes, 

Caitlin Moyer, Ph.D.

Associate Editor 

PLOS Medicine

plosmedicine.org